J Physiol 601.7 (2023) pp 1207–1224

# On the interdependence of ketone body oxidation, glycogen content, glycolysis and energy metabolism in the heart

Azrul Abdul Kadir[1] 🔲, Brianna J. Stubbs[2], Cher-Rin Chong[3], Henry Lee[1], Mark Cole[4] 🔲, Carolyn Carr[1] 🔲, David Hauton[5], James McCullagh[5] 🔲, Rhys D. Evans[1] 🔲 and Kieran Clarke[1]

[1] Department of Physiology, Anatomy and Genetics, University of Oxford, Oxford, UK
[2] Buck Institute for Research on Aging, Novato, CA, USA
[3] Faculty of Health and Medical Sciences, University of Adelaide, Adelaide, Australia
[4] School of Life Sciences, University of Nottingham, Nottingham, UK
[5] Department of Chemistry, University of Oxford, Oxford, UK

Handling Editors: Michael Hogan & Bettina Mittendorfer

The peer review history is available in the Supporting information section of this article
(https://doi.org/10.1113/JP284270#support-information-section).

*The Journal of Physiology*

**Azrul Abdul Kadir** completed his Bachelor's degree in biology from the University of Putra, Malaysia, his Master's in Molecular Medicine from the University of Sheffield, UK, and his Doctorate in Cardiac Metabolism from the University of Oxford, UK, supervised by Dr Rhys Evans and Prof. Kieran Clarke. He is currently a Research Fellow in the Cardiovascular Research Centre at Harvard Medical School, Boston, MA, working with Dr Anthony Rosenzweig. He represented the Cardiology Division of Mass General Hospital in the 2021—2022 MGH Leadership programme to build an impactful network and research. He plays squash in his spare time.

**Abstract**  In heart, glucose and glycolysis are important for anaplerosis and potentially therefore for D-$\beta$-hydroxybutyrate ($\beta$HB) oxidation. As a glucose store, glycogen may also furnish anaplerosis. We determined the effects of glycogen content on $\beta$HB oxidation and glycolytic rates, and their downstream effects on energetics, in the isolated rat heart. High glycogen (HG) and low glycogen (LG) containing hearts were perfused with 11 mM [5-$^3$H]glucose and/or 4 mM [$^{14}$C]$\beta$HB to measure glycolytic rates or $\beta$HB oxidation, respectively, then freeze-clamped for glycogen and metabolomic analyses. Free cytosolic [NAD$^+$]/[NADH] and mitochondrial [Q$^+$]/[QH$_2$] ratios were estimated using the lactate dehydrogenase and succinate dehydrogenase reaction, respectively. Phosphocreatine (PCr) and inorganic phosphate (P$_i$) concentrations were measured using $^{31}$P-nuclear magnetic resonance spectroscopy. Rates of $\beta$HB oxidation in LG hearts were half that in HG hearts, with $\beta$HB oxidation directly proportional to glycogen content. $\beta$HB oxidation decreased glycolysis in all hearts. Glycogenolysis in glycogen-replete hearts perfused with $\beta$HB alone was twice that of hearts perfused with $\beta$HB and glucose, which had significantly higher levels of the glycolytic intermediates fructose 1,6-bisphosphate and 3-phosphoglycerate, and higher free cytosolic [NAD$^+$]/[NADH]. $\beta$HB oxidation increased the Krebs cycle intermediates citrate, 2-oxoglutarate and succinate, the total NADP/H pool, reduced mitochondrial [Q$^+$]/[QH$_2$], and increased the calculated free energy of ATP hydrolysis ($\Delta G_{ATP}$). Although $\beta$HB oxidation inhibited glycolysis, glycolytic intermediates were not depleted, and cytosolic free NAD remained oxidised. $\beta$HB oxidation alone increased Krebs cycle intermediates, reduced mitochondrial Q and increased $\Delta G_{ATP}$. We conclude that glycogen facilitates cardiac $\beta$HB oxidation by anaplerosis.

(Received 14 December 2022; accepted after revision 23 January 2023; first published online 17 February 2023)

**Corresponding authors** A. A. Kadir and R. Evans: Department of Physiology, Anatomy and Genetics, Sherrington Building, University of Oxford, Parks Road, Oxford OX1 3PT, UK.    Email: rhys.evans@dpag.ox.ac.uk and azrulabdulkadir@gmail.com

**Abstract figure legend** Overview of relationship of glycogen to ketone body oxidation and cardiac energetics in isolated perfused rat hearts. Myocardial glycogen was pre-labelled with tritium and its metabolic fate tracked using a pulse–chase technique. Increased glycolytic flux from glycogen facilitated increased exogenous $\beta$-hydroxybutyrate ($\beta$HB) oxidation through anaplerosis, and the increased $\beta$HB oxidation increased mitochondrial redox potential, and hence increased free energy of ATP hydrolysis.

## Key points

- Ketone bodies (D-$\beta$-hydroxybutyrate, acetoacetate) are increasingly recognised as important cardiac energetic substrates, in both healthy and diseased hearts.
- As 2-carbon equivalents they are cataplerotic, causing depletion of Krebs cycle intermediates; therefore their utilisation requires anaplerotic supplementation, and intra-myocardial glycogen has been suggested as a potential anaplerotic source during ketone oxidation.
- It is demonstrated here that cardiac glycogen does indeed provide anaplerotic substrate to facilitate $\beta$-hydroxybutyrate oxidation in isolated perfused rat heart, and this contribution was quantified using a novel pulse–chase metabolic approach.
- Further, using metabolomics and $^{31}$P-MR, it was shown that glycolytic flux from myocardial glycogen increased the heart's ability to oxidise $\beta$HB, and $\beta$HB oxidation increased the mitochondrial redox potential, ultimately increasing the free energy of ATP hydrolysis.

## Introduction

Ketone bodies (KB; D-$\beta$-hydroxybutyrate, acetoacetate) are energy-providing substrates produced physiologically during starvation. They are synthesised from fatty acids in the liver, the fatty acids being derived from lipolysis of triacylglycerol stored in adipose tissue. Functionally, they are a transport form of acetyl-CoA, and constitute a glucose-sparing resource in catabolic states. They are utilised by all oxidative tissues, notably brain and muscle, except liver. However, whilst they spare some glucose utilisation in starvation (and hence permit only

limited glucose storage, in the form of glycogen, in anabolic states), they have a complicated relationship with carbohydrate availability, since their metabolism requires additional metabolic provision in the form of carbohydrates and/or proteins (anaplerosis). Unlike the situation in a ketogenic (low carbohydrate) diet, exogenous ketones delivered by intravenous infusion (Balasse & Ooms, 1968; Mikkelsen et al., 2015) or in ketone drinks (Clarke et al., 2012; Shivva et al., 2016) can raise blood D-β-hydroxybutyrate (βHB) concentrations in the presence of normal, or even high, glucose levels. In the glycogen replete (fed) state, exogenous βHB oxidation following a ketone ester drink may account for ~10–18% of total oxygen consumption during exercise in humans (Cox et al., 2016), although βHB oxidation rate in cardiac muscle under these conditions is uncertain.

Since two carbons are lost in the Krebs (tricarboxylic acid) cycle ($2 \times CO_2$), acetyl-CoA, the dedicated 2-carbon substrate of the cycle, is unable to replenish the cycle components (all $\geq$4 carbons). Furthermore, acetyl-CoA cannot be carboxylated to pyruvate, a potential source of four carbon units, because pyruvate dehydrogenase (PDH) is far from equilibrium and effectively irreversible. Hence, oxidation of two carbon units in the form of acetyl-CoA, derived from fatty acids or ketone bodies, is cataplerotic, leading to depletion of Krebs cycle intermediates. As captured in the aphorism 'fat burns in the fire of carbohydrate' (Frayn & Evans, 2019), carbohydrate (or other sources of $\geq$3-carbon substrate such as glucogenic amino acids) is therefore required to maintain Krebs cycle intermediates and facilitate fatty acid oxidation, so may also be required to oxidise ketone bodies (Taegtmeyer, 1983; Taegtmeyer et al., 1980). Glucose and βHB are both metabolised to 2-carbon acetyl-CoA, which requires 4-carbon oxaloacetate to form 6-carbon citrate in the Krebs cycle (Fig. 1). However, unlike KBs and fatty acids, glucose – via 3-carbon pyruvate and pyruvate carboxylase – can also form oxaloacetate, hence maintaining levels of Krebs cycle intermediates and enabling acetyl-CoA from βHB to enter the Krebs cycle for oxidation – the process of anaplerosis (Russell & Taegtmeyer, 1991a, 1991b, 1992; Russell et al., 1997; Taegtmeyer, 1983; Taegtmeyer et al., 1980).

Glycogen occupies about 2% of the cell volume in the adult cardiac myocyte, the storage capacity for glycogen being limited because it is stored with water of hydration (Kassiotis et al., 2008). Myocardial glycogen does not liberate glucose during normal cardiac activity (Goodwin et al., 1995), but is reserved to serve as an energy 'buffer' to maintain endogenous glucose provision and metabolism under conditions of energetic stress (Evans, 1934; Rabinowitz, 1971). Hence it acts as both an energetic reserve, and also, potentially, an anaplerotic reserve. During brief, low-flow ischaemia, glycogen prevents myocardial injury by maintaining sufficient glycolytic ATP

for the sarcolemmal Na$^+$ pump to keep intracellular Na$^+$ low, despite increased Na$^+$/H$^+$ exchange; however, during prolonged ischaemia when glycogen is fully depleted, myocardial injury occurs due to ATP depletion leading to low pH and high intracellular [Na$^+$] (Cross et al., 1996; Taegtmeyer, 2004).

We hypothesised that myocardial glycogen may specifically contribute, via glycolysis to pyruvate, to the intermediary compounds required for anaplerosis and hence the efficient oxidation of βHB in the working myocardium and maintenance of myocardial energetic status (Russell & Taegtmeyer, 1991a, 1991b, 1992; Russell et al., 1997; Taegtmeyer, 1983; Taegtmeyer et al., 1980). Here, in the isolated perfused rat heart, we determined the effects of glycogen levels on βHB oxidation and glycolytic rates under conditions of defined exogenous

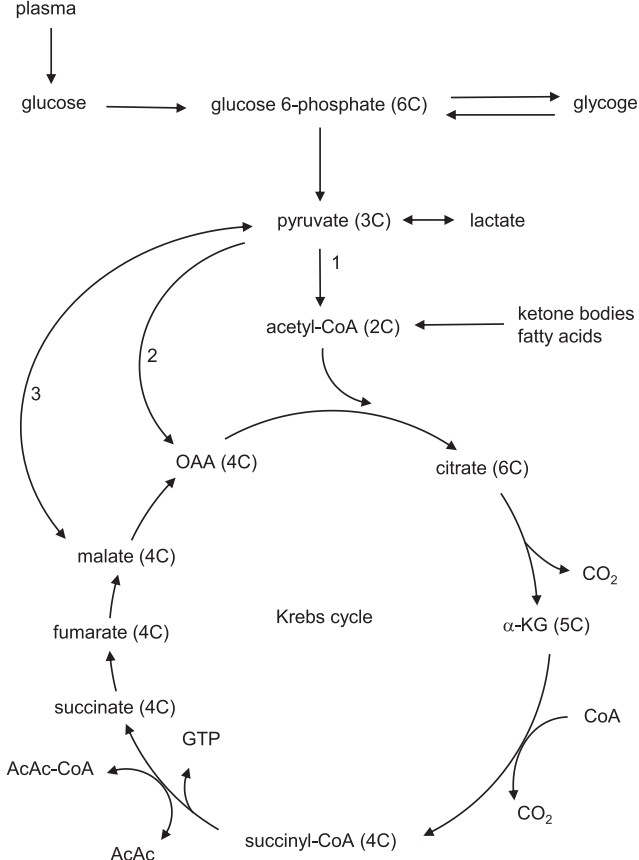

**Figure 1. Glucose and glycogen metabolism for anaplerosis**
Numbers refer to the enzymes catalysing the specific reactions. 3-Carbon (3C) units are required for anaplerosis; 2-carbon (2C) units are cataplerotic. Pyruvate (3C), derived from glucose or glycogen, can be decarboxylated to acetyl-CoA (2C) by PDH (1), carboxylated to oxaloacetate (4C) by pyruvate carboxylase (2) and converted to malate (4C) by malic enzyme (3). AcAc, acetoacetate; AcAc-CoA, acetoacetyl-CoA; α-KG, α-ketoglutarate (2-oxoglutarate); CoA, coenzyme A; GTP, guanosine triphosphate; OAA, oxaloacetate.

substrate availability, together with a novel estimation of glycogen-derived anaplerotic flux, intermediates of glycolysis and the Krebs cycle, and accompanying energetics.

## Methods

### Ethical approval

Experiments were performed in accordance with the UK Home Office guidelines under The Animals (Scientific Procedures) Act 1986, under Home Office Project Licence PPL number PP9434487, and under the guidelines of the University of Oxford Animal Welfare Ethical Review Board.

### Animals

Male adult Wistar rats (200–250 g) were obtained from a commercial breeder (Harlan, Bicester, UK) and maintained at 20 ± 2°C ambient temperature on a standard laboratory chow diet (comprising by weight approximately 52% carbohydrate, 21% protein and 4% fat; the residue was non-digestible material (Special Diet Services, Witham, UK)) with a 12 h light–12 h dark cycle and free access to water.

### Chemicals

Radiochemicals were obtained from American Radio-labelled Chemicals Inc., St Louis, MO, USA; other biochemicals (AnalaR grade) were obtained from Sigma/Merck Life Science UK Ltd, Gillingham, UK.

### Perfusion of isolated rat hearts

Isolated hearts from fed rats were perfused in the Langendorff mode. Following terminal anaesthesia with sodium pentobarbital (60 mg kg$^{-1}$ body weight by intraperitoneal injection), general anaesthesia was confirmed, then thoracotomy performed and hearts were rapidly excised and placed in ice-cold Krebs–Henseleit (KH) buffer. Hearts were cannulated via the aorta and perfused retrogradely at a constant perfusion pressure of 100 mmHg at 37°C. Hearts were perfused with 200 ml recirculating KH buffer (mM: 118 NaCl, 4.7 KCl, 1.2 MgSO$_4$, 1.3 CaCl$_2$, 0.5 EDTA, 25 NaHCO$_3$, 1.2 KH$_2$PO$_4$, pH 7.4) together with substrates as detailed, and gassed with 95% O$_2$ and 5% CO$_2$.

To measure cardiac function during the perfusion protocol, a fluid filled PVC balloon attached via a poly-ethylene tube to a bridge amplifier (ADInstruments, Oxford, UK) and PowerLab data acquisition system was inserted into the left ventricle and inflated to 4−6 mmHg

left ventricular end-diastolic pressure. Left ventricular developed pressure (DP) was determined as peak systolic pressure minus end-diastolic pressure. Rate pressure product (RPP) was calculated as the product of developed pressure and heart rate (HR).

### Measurement of myocardial D-$\beta$-hydroxybutyrate oxidation, glycolysis and glycogenolysis

Hearts were initially perfused with oxygenated, non-recirculating KH buffer to either decrease or increase the glycogen content. To deplete glycogen, hearts were perfused with substrate (glucose)-free KH buffer for 20 min (low glycogen: LG) (Cross et al., 1996; Goodwin et al., 1996); and to augment glycogen content, hearts were perfused with KH buffer containing pyruvate (4 mM), lactate (1 mM), glucose (11 mM) and insulin (50 mU min$^{-1}$) for 60 min (high glycogen: HG) (Cross et al., 1996). In initial experiments, hearts perfused under these conditions were freeze-clamped in order to confirm the effect of substrate manipulation on myocardial glycogen content.

For the measurement of $\beta$HB oxidation rates, LG and HG hearts were perfused with KH buffer containing 11 mM glucose in the presence (LG +$\beta$HB; HG +$\beta$HB) or absence (LG −$\beta$HB; HG −$\beta$HB) of 4 mM D-3-hydroxybutyrate (sodium salt) supplemented with 5 $\mu$Ci D-[1-$^{14}$C]-3-hydroxybutyrate (sodium salt); this concentration was chosen to reflect both the fasting state and conditions of exogenous ketone administration. $\beta$HB oxidation rates were determined by collection of the $^{14}$CO$_2$ produced (combined gaseous $^{14}$CO$_2$ collection and $^{14}$CO$_2$ liberated from perfusate H$^{14}$CO$_3{}^-$) at 10-min intervals for 40 min, as described (Hauton et al., 2001). For the measurement of glycolytic rates, buffer was supplemented with 0.2 $\mu$Ci [5-$^3$H]glucose plus 11 mM glucose, with or without 4 mM $\beta$HB (LG/HG $\pm\beta$HB) (Lopaschuk & Barr, 1997). The rates of glycolysis were determined using the production of $^3$H$_2$O collected at 4-min intervals for 40 min – perfusate aliquots were subsequently passed through Dowex ion-exchange resin columns to separate labelled glucose from labelled water. Rates of glycogenesis were determined by the difference in myocardial glycogen content.

In this study, we hypothesised that glycogen can be utilised for anaplerosis to oxaloacetate, to increase the heart's ability to oxidise $\beta$HB. Direct, accurate, measurement of oxaloacetate synthesis from glycogen is technically challenging because oxaloacetate is present in concentrations of only 5−10 nmol/g and its turnover rate is up to 800 times per minute (Chung et al., 2015; Des Rosiers et al., 2011). Therefore, a novel technique was devised and a modified perfusion protocol was performed to determine glycogenolysis,

and hence glycogen contribution to the myocardial carbohydrate pool, in high glycogen content (HG) hearts perfused with or without $\beta$HB, with anaplerosis estimated indirectly from labelled glycogen mobilisation using a pulse–chase labelling technique. Hearts were initially perfused with glucose-free KH buffer for 20 min to deplete glycogen (see above) (Cross et al., 1996; Goodwin et al., 1996); subsequently, for the 'pulse' experiment, the hearts were then perfused with KH buffer containing [5-$^3$H]glucose, glucose (11 mM), pyruvate (4 mM), lactate (1 mM), insulin (50 mU min$^{-1}$) and $\beta$HB (4 mM) for 60 min to replenish and incorporate [5-$^3$H]glucose into [$^3$H]glycogen (pulse) (high glycogen). Glycogenolysis from this [$^3$H]glycogen in HG hearts was then determined during the 'chase' that contained either substrate-free buffer ($-$Gluc $-\beta$HB), 11 mM glucose alone ($+$Gluc $-\beta$HB), 4 mM $\beta$HB alone ($-$Gluc $+\beta$HB), or 11 mM glucose plus 4 mM $\beta$HB ($+$Gluc $+\beta$HB). The production of $^3$H$_2$O, collected at 4-min intervals for 40 min, was used to determine the rate of glycogenolysis from $^3$[H]glycogen (chase), hence enabling glycogen contribution to anaplerosis to be measured.

We reasoned that in glycogen-replete hearts perfused with ample exogenous glucose and $\beta$HB as energetic substrate (and ignoring any contribution to the pentose phosphate pathway – known to be a quantitatively relatively minor pathway in heart (Zimmer, 1992; Zimmer et al., 1984), endogenous myocardial glycogen would not be required for energetic provision and any residual glycogen utilisation (glycogenolysis) would likely be destined for anaplerosis. We further reasoned that when hearts were perfused with (cataplerotic) ketone body alone, PDH inhibition would prevent a significant contribution of glycogen-derived pyruvate to energy production (AcCoA synthesis) but would further increase anaplerotic requirement, with endogenous glycogen as the sole source – hence anaplerosis from glycogen would increase to facilitate $\beta$HB oxidation. By contrast, hearts perfused without any exogenous substrate would rely on high rates of glycogenolysis from endogenous glycogen providing both anaplerosis and oxidative substrate. This is represented in Fig. 2.

Hearts were freeze-clamped in liquid N$_2$-cooled Wollenberger tongs (Wollenberger et al., 1973) at the end of the perfusion for glycogen and metabolomic analyses. Myocardial glycogen content was determined in ground frozen tissue using a spectrophotometric enzymatic-coupled assay as described (Bergmeyer, 1974).

## Metabolomic analyses

Hearts were freeze-clamped at the end of perfusion for metabolomic analyses. Briefly, heart tissue (0.1 g) was extracted in 80% methanol and homogenised using a Precellys homogeniser (Le Belle et al., 2002). The sample was centrifuged at 23,000 $g$ for 20 min. Each sample was diluted to create a final DNA concentration of 100 ng $\mu$l$^{-1}$. The samples were analysed using up to three separate liquid chromatography (LC) coupled to mass spectrometry (LC–MS) methods using two different LC systems (Thermo Scientific ICS-5000+

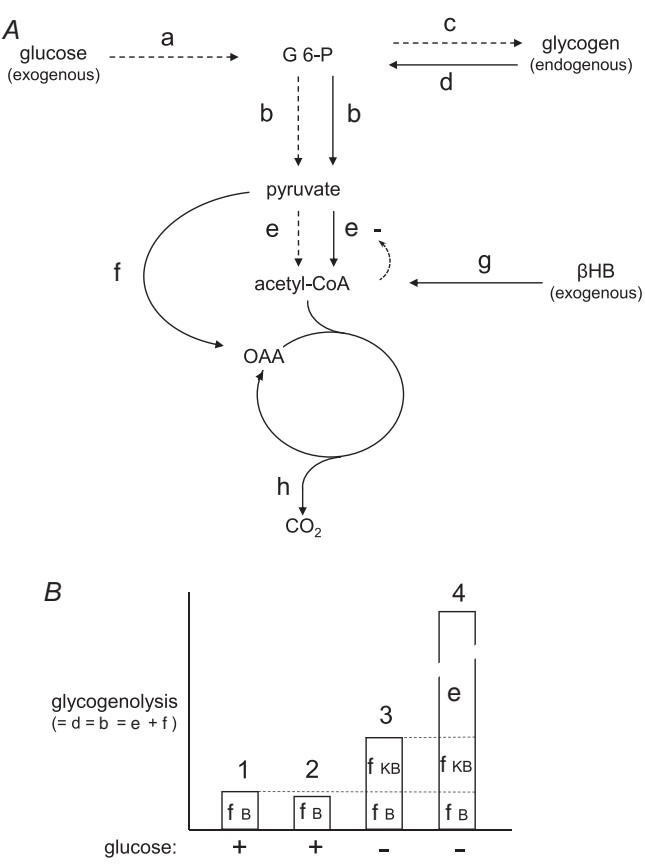

**Figure 2. Metabolic flux of exogenous (glucose) and endogenous (glycogen) carbohydrate contributing to anaplerosis**

*A*, glucose is assimilated into the cell (glucose uptake: *a*) and then undergoes glycolysis (*b*) or glycogen synthesis (glycogenesis: *c*). Glycogen undergoes glycogenolysis (*d*) and subsequent glycolysis (*b'*). The resulting pyruvate can undergo oxidative decarboxylation (*e* from glucose; *e'* from glycogen), resulting in cataplerotic (2C) acetyl-CoA, or anaplerosis (*f'* from glycogen). Also shown is (cataplerotic) $\beta$-hydroxybutyrate ($\beta$HB) utilisation (*g*), and total oxidation (*h*). *B*, assuming no other metabolic fates, glycogenolytic flux (*d*) is identical to glycogen-derived glycolysis (*b'*), which is also the sum of pyruvate oxidation (*e'*) and anaplerosis (*f'*). If oxidation of glycogen-derived pyruvate is negligible (e.g. when abundant exogenous substrate, such as glucose and $\beta$HB, is available) then glycogenolysis (*d*) will equal anaplerosis (*f'*). If alternative anaplerotic sources are available (e.g. exogenous glucose) then anaplerosis from glycogen will be low – basal *f'* (*f'*$_B$); however, if glycogen is the sole anaplerotic resource, increased anaplerosis from glycogen will result: the additional anaplerosis from glycogen to facilitate $\beta$HB oxidation is designated *f'*$_{KB}$. For further details see text.

ion chromatography system and a Thermo Ultimate 3000; Thermo Scientific, San Jose, CA, USA) and appropriate standards. The LC–MS methods were ion exchange chromatography–mass spectrometry (IC-MS), C18 reversed phase (underivatised) and C18 reversed phase (derivatised). Each was coupled directly to a Q-Exactive HF Hybrid Quadrupole-Orbitrap mass spectrometer with a HESI II electrospray ionisation source (Thermo Scientific). Raw data files were processed using ProgenesisQI (Waters, Elstree, UK). Retention times, accurate mass values, relative isotope abundances and fragmentation patterns were compared between authentic standards and the samples measured. Data (absolute ion intensities) are expressed in arbitrary units as the ratio of metabolite peak relative to the internal standard. Identifications were accepted only when the following criteria were met: <5 ppm differences between measured and theoretical mass (based on chemical formula), <30 s differences between authentic standard and analyte retention times, isotope peak abundance measurements for analytes were >90% matched to the theoretical value generated from the chemical formula. Fragmentation patterns were matched to the lowest base peak and two additional peak matches in the MS spectrum to within 12 ppm. In total, 68 metabolites were identified and quantified per heart.

### Cytosolic redox state

The free cytosolic $[NAD^+]/[NADH]$ ratio was estimated from the measured components of the lactate dehydrogenase reaction. The substrates of lactate dehydrogenase are likely to be in equilibrium with free $NAD^+$ and NADH, so the ratio of the free dinucleotides can be calculated from the measured concentrations of the substrates (Veech et al., 1969; Williamson et al., 1967):

$$\frac{[NAD^+]}{[NADH]} = \frac{[Lactate]}{[Pyruvate]}$$

where lactate and pyruvate were measured using metabolomic analysis.

### Mitochondrial Q couple

The free mitochondrial $[Q^+]/[QH_2]$ ratio was estimated from the measured components of the succinate dehydrogenase reaction. The substrates of succinate dehydrogenase are likely to be in equilibrium with free $Q^+$ and $QH_2$, and the ratio of the free mitochondrial $[Q^+]/[QH_2]$ can be calculated from the measured concentrations of the substrates (Sato et al., 1995; Veech et al., 1969; Williamson et al., 1967):

$$\frac{[Q^+]}{[QH_2]} = \frac{[Fumarate]}{[Succinate]}$$

where fumarate and succinate were measured using metabolomic analysis.

### Cardiac energetics measured using nuclear magnetic resonance spectroscopy

To measure cytosolic ATP, PCr and $P_i$, a fully relaxed spectrum was acquired for each heart using a 500 MHz (11.7 T) magenetic resonance system, comprising an 11.7 T vertical-bore magnet (Magnex Scientific, Oxford, UK) and a Bruker Avance console (Bruker Medical, Ettlingen, Germany) as previously described (Cross et al., 1996; Dodd et al., 2015). Nuclear magnetic resonance (NMR) spectra were analysed in the time domain using the AMARES algorithm in the jMRUI 5.2 software package (MRUI Consortium, Universitat Autònoma de Barcelona, Barcelona, Spain).

### Statistical analysis

Statistical analysis was performed using GraphPad Prism version 8.1.1 for Mac OS X, GraphPad Software, San Diego, CA, USA. The effect of substrate manipulation on left ventricular function and metabolic parameters were compared by two-way analysis of variance (ANOVA), with glycogen content/glucose and $\beta$HB as factors, and Student's unpaired $t$ test, as appropriate. The effect of $\beta$HB oxidation on Krebs cycle intermediates measured by metabolomics were also compared using two-way ANOVA. Tukey *post hoc* corrections were made for multiple comparisons to identify specific significant differences. Number of observations ($n$) is the number of individual heart perfusions. Results, presented as means $\pm$ SD ($n$) unless otherwise specified, were considered significant at $P < 0.05$.

## Results

### The effect of different substrate availability on left ventricular function

Generally, the cardiodynamics of perfused hearts during substrate manipulation was stable over the perfusion protocol (Table 1). The perfusion of hearts with KH buffer without substrate in order to deplete glycogen (low glycogen) led to a small decrease in heart rate by the end of the perfusion period compared to heart perfused with buffer containing pyruvate + lactate + glucose + insulin in order to increase glycogen levels (high glycogen), but HR normalised once substrate was re-introduced into the perfusate (Table 1). Perfusion of high glycogen hearts with the addition of $\beta$HB was associated with a small but significant decrease in DP; however the recorded rate–pressure product (RPP; HR × DP), a composite

**Table 1. Cardiodynamics and left ventricular function in isolated perfused rat hearts following glycogen manipulation, perfusion with glucose alone, or perfusion with glucose plus βHB**

| | Heart rate (min$^{-1}$) | | Developed pressure (mmHg) | | Rate-pressure product (mmHg min$^{-1}$) | |
|---|---|---|---|---|---|---|
| | Low glycogen | High glycogen | Low glycogen | High glycogen | Low glycogen | High glycogen |
| After glycogen manipulation (A) | 229 ± 40 (9) | 280 ± 35 (5) | 90 ± 10 (9) | 84 ± 9 (5) | 20 706 ± 5031(9) | 23 567 ± 4800 (5) |
| After perfusion with glucose alone (B) | 241 ± 41 (7) | 269 ± 48 (7) | 99 ± 23 (7) | 77 ± 29 (7) | 23 078 ± 3429 (7) | 21 143 ± 9264 (7) |
| After perfusion with glucose + βHB (C) | 311 ± 233 (9) | 285 ± 40 (5) | 89 ± 33 (9) | 67 ± 6 (5) | 22 701 ± 5861 (9) | 19 272 ± 3746 (5) |
| *P* | | | | | | |
| A *vs.* B | 0.580 | 0.686 | 0.312 | 0.650 | 0.304 | 0.607 |
| A *vs.* C | 0.445 | 0.819 | 0.949 | 0.008 | 0.450 | 0.153 |
| B *vs.* C | 0.899 | 0.553 | 0.526 | 0.458 | 0.882 | 0.681 |
| A: LG *vs.* HG | | 0.036 | | 0.297 | | 0.321 |
| B: LG *vs.* HG | | 0.250 | | 0.151 | | 0.614 |
| C: LG *vs.* HG | | 0.818 | | 0.182 | | 0.264 |

Values are means ± SD (*n* hearts). Left ventricular function, recorded during the entire perfusion period, is shown as heart rate (HR), left ventricular developed pressure (DP; peak left ventricular systolic pressure – left ventricular end-diastolic pressure) and left ventricular rate–pressure product (calculated as HR × DP). For further details see text. βHB: D-β-hydroxybutyrate.

measure of left ventricular function, was not significantly altered in these hearts – indeed, there was no significant difference in RPP between any of the experimental groups (low glycogen or high glycogen; perfused with glucose alone or perfused with glucose plus βHB) throughout the respective perfusion protocols (Table 1).

### The effect of myocardial glycogen content on βHB oxidation

Pre-perfusion of hearts with different substrate complements was found to alter residual myocardial glycogen content. Thus, pre-perfusion with substrate-free buffer was confirmed to decrease myocardial glycogen content whilst pre-perfusion with high substrate content perfusate augmented cardiac glycogen. Following an initial glycogen-depleting (substrate-free) phase, cardiac glycogen content was 5.3 μmol glycosyl units g wet weight (ww)$^{-1}$. However, following perfusion of hearts with substrate-rich perfusate, cardiac glycogen levels were found to be augmented to 43.5 μmol glycosyl units gww$^{-1}$ (Fig. 3*A*).

In low glycogen (LG) hearts, subsequent perfusion with glucose alone (+Gluc −βHB) increased glycogen content moderately – by about three times – but replete levels of glycogen were not achieved: exogenous glucose, being the only exogenous oxidative substrate source under these conditions and despite apparently adequate concentration (11 mM), presumably was insufficient to support cardiac oxidative metabolism and also fully replete glycogen content. However, high glycogen (HG) hearts perfused with glucose alone (+Gluc −βHB) had

fully replete glycogen content at the end of the perfusion (i.e. significantly more than hearts starting with low glycogen; Fig. 3*A*). This suggests that exogenous glucose was capable of fully supporting cardiac metabolism with replete glycogen stores, but was not able to fully replete glycogen stores under these conditions and in this time frame. By contrast, LG heart perfused with both glucose and βHB (+Gluc +βHB) had significantly greater increased glycogen content at the end of the perfusion (about seven-fold, from 5.3 to 37.1 μmol glycosyl units gww$^{-1}$). This glycogen content was likely maximal, since HG hearts perfused with both substrates (+Gluc +βHB) did not have significantly further increased glycogen content at the end of the perfusion (Fig. 3*A*). These data suggest that the presence of βHB as an oxidative substrate in LG hearts spared some exogenous glucose from oxidation and hence was directed to glycogen synthesis.

βHB oxidation rates in LG hearts were about half that seen in HG hearts (0.72 ± 0.14 *vs.* 1.41 ± 0.25 μmol gww$^{-1}$ min$^{-1}$; $P = 0.025$; Fig. 3*B*). Furthermore, there was a positive correlation between βHB oxidation rate and myocardial glycogen content at the end of perfusion ($R^2 = 0.51$, $P = 0.019$; Fig. 3*C*).

### The effect of βHB oxidation and glycogen on glucose utilisation

The net rate of glycogenesis was measured using the difference in glycogen content before and following the 40 min perfusion period (Table 2; Fig. 4*A*). In LG hearts, perfusion with glucose alone was associated with relatively

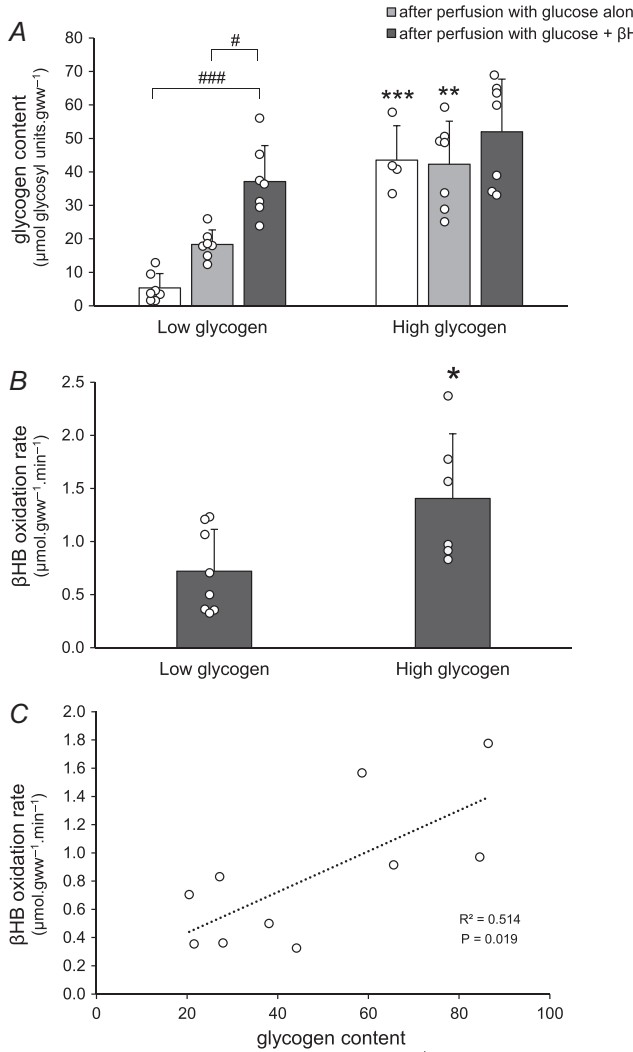

**Figure 3. Glycogen content and D-β-hydroxybutyrate (βHB) oxidation rates in isolated perfused rat heart**
*A*, myocardial glycogen content was measured after altering glycogen levels (glycogen manipulation), confirming depleted and augmented tissue levels; subsequently (following glycogen manipulation) hearts were perfused with glucose alone, or with glucose plus βHB, and glycogen content again measured. Data analysis by ANOVA and Tukey correction: *P*: LG am *vs.* HG am: <0.0001; LG Gluc *vs.* HG Gluc: 0.002; LG βHB *vs.* HG βHB: 0.123; LG am *vs.* LG Gluc: 0.227; LG am *vs.* LG βHB: <0.0001; LG Gluc *vs.* LG βHB: 0.026; HG am *vs.* HG Gluc: 0.989; HG am *vs.* HG βHB: 0.801; HG Gluc *vs.* HG βHB: 0.537. *B*, βHB oxidation rates in low and high glycogen hearts after perfusion with glucose plus βHB. Data analysis by unpaired *t* test: *P* = 0.025. *C*, myocardial βHB oxidation rates plotted against the final myocardial glycogen content in all hearts after subsequent perfusion with glucose plus βHB. For further details see text. Data analysis of linear correlation by Pearson correlation coefficient; two tailed *P* = 0.019. Values are means ± SD. Effect of glycogen content is indicated as: *$*P < 0.05$, $**P < 0.01$, $***P < 0.001$. Effect of βHB is indicated as: #$P < 0.05$, ###$P < 0.001$. Significance was taken at $P < 0.05$. am, after glycogen manipulation; βHB, after perfusion with glucose + βHB; Gluc, after perfusion with glucose alone; HG, high glycogen; LG, low glycogen.

moderate rates of glycogenesis ($0.31 \pm 0.04$ μmol glycosyl units gww$^{-1}$ min$^{-1}$); addition of βHB to the glucose (+βHB) markedly increased glycogenesis to $0.80 \pm 0.09$ μmol glycosyl units gww$^{-1}$ min$^{-1}$ (Table 2; Fig. 4*A*), indicating sparing of exogenous glucose utilisation for glycogen synthesis and reflecting the accumulation of glycogen content (see Fig. 3*A*). By contrast, rates of glycogenesis in HG hearts were lower than in LG hearts, as expected, since glycogen stores were already present in the HG state: glycogen-replete hearts perfused with glucose alone (−βHB) showed zero net glycogenesis, although again addition of βHB to glucose in the perfusate increased glycogenesis modestly but significantly to $0.21 \pm 0.04$ (μmol glycosyl units gww$^{-1}$ min$^{-1}$; Table 2; Fig. 4*A*).

Glycolysis from exogenous glucose in HG hearts perfused with glucose alone (−βHB) was significantly higher than that of the LG hearts ($0.77 \pm 0.03$ *vs.* $0.50 \pm 0.08$ μmol gww$^{-1}$ min$^{-1}$, $P < 0.05$). Addition of βHB significantly decreased glycolysis to 0.22 and 0.24 μmol gww$^{-1}$ min$^{-1}$ in low and high glycogen content hearts, respectively (Table 2; Fig. 4*B*). Hence a reciprocal relationship between metabolic fate of exogenous glucose into myocardial glycogen stores or into glycolysis was demonstrated.

Rates of exogenous glucose utilisation were calculated as the sum of total glycolytic and glycogenic rates. Rates of glucose utilisation in hearts perfused with glucose alone (−βHB) were high, and similar, regardless of tissue glycogen status (Table 2). In HG hearts, addition of βHB (+βHB), with accompanying high rates of βHB oxidation ($1.41 \pm 0.25$ μmol gww$^{-1}$ min$^{-1}$; Fig. 3*B*), significantly decreased rates of exogenous glucose utilisation by about 40% (from $0.77 \pm 0.03$ to $0.45 \pm 0.06$ μmol gww$^{-1}$ min$^{-1}$, i.e. −0.32 μmol gww$^{-1}$ min$^{-1}$). However, in LG hearts, addition of βHB (+βHB) further increased rates of exogenous glucose utilisation by about 25% (from $0.81 \pm 0.12$ to $1.01 \pm 0.08$ μmol gww$^{-1}$ min$^{-1}$, i.e. +0.2 μmol gww$^{-1}$ min$^{-1}$), accompanied by relatively low rates of βHB oxidation ($0.72 \pm 0.14$ μmol gww$^{-1}$ min$^{-1}$; Fig. 3*B*). These effects of βHB oxidation rates on exogenous glucose utilisation in low and high glycogen content hearts are summarised in Table 2.

## Myocardial glycogenesis and glycogenolysis from ³H-labelled glycogen in HG hearts

Glycogen may provide anaplerotic substrates, via glycolysis to pyruvate, to increase the heart's ability to oxidise βHB. Direct measurement of anaplerosis from glycogen is problematic, and therefore a modified perfusion protocol using a pulse–chase labelling technique was devised in order to determine glycogenolysis in high glycogen (HG) hearts perfused

**Table 2.** D-*β*-Hydroxybutyrate oxidation and glucose metabolism in low and high glycogen content rat hearts

| | Low glycogen | | High glycogen | |
|---|---|---|---|---|
| | $-\beta$HB | $+\beta$HB | $-\beta$HB | $+\beta$HB |
| $\beta$HB oxidation rate ($\mu$mol gww$^{-1}$ min$^{-1}$) (A) | — | 0.72 ± 0.39 (8) | — | 1.41 ± 0.69* (6) |
| Glycolysis ($\mu$mol gww$^{-1}$ min$^{-1}$) (B) | 0.50 ± 0.21 (7) | 0.22 ± 0.17# (7) | 0.77 ± 0.09* (7) | 0.24 ± 0.14### (7) |
| Net glycogenesis ($\mu$mol gww$^{-1}$ min$^{-1}$) (C) | 0.31 ± 0.10 (7) | 0.80 ± 0.25### (7) | 0.0 ± 0.0*** (7) | 0.21 ± 0.10***### (7) |
| Exogenous glucose utilisation ($\mu$mol gww$^{-1}$ min$^{-1}$) (D) | 0.81 ± 0.33 (7) | 1.01 ± 0.21 (7) | 0.77 ± 0.09 (7) | 0.46 ± 0.16***### (7) |
| *P* | | | | |
| A: LG *vs*. HG | | | | 0.025 |
| B: LG *vs*. HG | | | 0.019 | 0.769 |
| C: LG *vs*. HG | | | <0.0001 | <0.0001 |
| D: LG *vs*. HG | | | 0.737 | 0.0001 |
| B: $+\beta$HB *vs*. $-\beta$HB | | 0.017 | | <0.0001 |
| C: $+\beta$HB *vs*. $-\beta$HB | | 0.0005 | | <0.0001 |
| D: $+\beta$HB *vs*. $-\beta$HB | | 0.200 | | 0.0007 |

Values are means ± SD (*n* hearts). Rat hearts replete with, or depleted of glycogen, were perfused with buffer containing either 11 mM glucose alone ($-\beta$HB), or 11 mM glucose plus 4 mM $\beta$HB ($+\beta$HB); for further details see text. Effect of glycogen is indicated: *$P < 0.05$, **$P < 0.01$, ***$P < 0.001$. Effect of $\beta$HB is indicated: #$P < 0.05$, ###$P < 0.001$. $\beta$HB: D-*β*-hydroxybutyrate.

with and without $\beta$HB, and hence indirectly estimate anaplerosis from labelled glycogen. Briefly, hearts were pre-labelled with [³H]glycogen (pulse), then glycogenolysis was measured in these glycogen-replete HG hearts subsequently perfused with and without $\beta$HB and/or glucose (chase); for further details, and explanation of the technique to assess rates of anaplerosis, see Methods above.

As expected, perfusion of these glycogen-replete hearts with both exogenous glucose and $\beta$HB (+Gluc +$\beta$HB) was associated with low rates of glycogenolysis (0.014 ± 0.001 $\mu$mol glucosyl units gww$^{-1}$ min$^{-1}$; Fig 5*A*), and high residual glycogen content following the perfusion (62.7 ± 3.3 $\mu$mol glucosyl units gww$^{-1}$; Fig 5*B*). This 'basal' rate of glycogenolysis likely represents minimal or zero contribution of glycogen to oxidative metabolism in the fully energy-replete and provisioned heart, but some constitutive glycogen-derived basal anaplerotic flux ($e'$ and $f'$ ($f'$B), respectively, in Fig. 2) exists. Perfusion of glycogen-replete hearts with exogenous glucose alone in the absence of $\beta$HB (+Gluc –$\beta$HB) was also associated with a similarly low rate of glycogenolysis and maintenance of a high residual glycogen content, as expected (Fig. 5), again suggesting adequate energy provision with low cataplerotic burden and hence low basal anaplerotic rate. However, perfusion with $\beta$HB but without glucose (–Gluc +$\beta$HB) – an enhanced cataplerotic burden – doubled the rate of glycogenolysis (to 0.026 ± 0.004 $\mu$mol glucosyl units gww$^{-1}$ min$^{-1}$; $P = 0.026$) and this was accompanied by a significant fall in glycogen content at the end of the perfusion (49.8 ± 4.4 $\mu$mol glucosyl units gww$^{-1}$; Fig. 5*A*);

this increased glycogenolysis represents increased anaplerotic flux to maintain Krebs cycle intermediates in the face of $\beta$HB-induced cataplerosis, with exogenous glucose no longer available for exogenous anaplerosis (see Fig 2*B* column 3: $f'$B + $f'_{KB}$): inevitably this leads to glycogen depletion. These data suggest an additional contribution of glycogen-derived anaplerotic flux of 0.026 – 0.014 = 0.012 $\mu$mol glucosyl units gww$^{-1}$ to maintain exogenous ketone metabolism: about 10% of maximal potential glycogenolytic flux. When glycogen-loaded hearts were perfused without exogenous substrate (–Gluc –$\beta$HB), endogenous glycogen was both the sole energy and anaplerotic source (Fig. 2*A*: $d = b' = e' + f'$; Fig. 2*B* column 4), and consequently the rate of glycogenolysis was maximal (0.102 ± 0.008 $\mu$mol glucosyl units gww$^{-1}$ min$^{-1}$), and residual glycogen content significantly lowered (36.9 ± $\mu$mol glucosyl units gww$^{-1}$; Fig. 5*B*), though still appreciable (about half-maximal) after this chase perfusion period (40 min), emphasising the ability of myocardial glycogen to maintain metabolic function as well as cardiodynamic function over this relatively extended period (see Table 1).

### The effect of $\beta$HB oxidation on glycolytic and Krebs cycle intermediates

Targeted metabolomic analysis, a method that prioritises compound identification and quantification, was used to assess the influence of $\beta$HB utilisation on Krebs cycle intermediates (Fig. 6). Glycogen-replete hearts were perfused with KH buffer containing no

substrate (−Gluc −$\beta$HB), 11 mM glucose alone (+Gluc −$\beta$HB), 4 mM $\beta$HB alone (−Gluc +$\beta$HB), or 11 mM glucose plus 4 mM $\beta$HB (+Gluc +$\beta$HB). Perfusion of glycogen-replete hearts with substrate-free buffer (−Gluc −$\beta$HB) increased fumarate, which was associated with decreased succinate. Thus, the free mitochondrial [$Q^+$]/[$QH_2$], calculated from the [fumarate]/[succinate] ratio (Veech et al., 1969; Williamson et al., 1967), indicated Q was highly oxidised in these hearts. In glycogen-replete hearts perfused with glucose alone (+Gluc −$\beta$HB), glucose significantly increased myocardial sorbitol, glycerol 3-phosphate, lactate, and hence

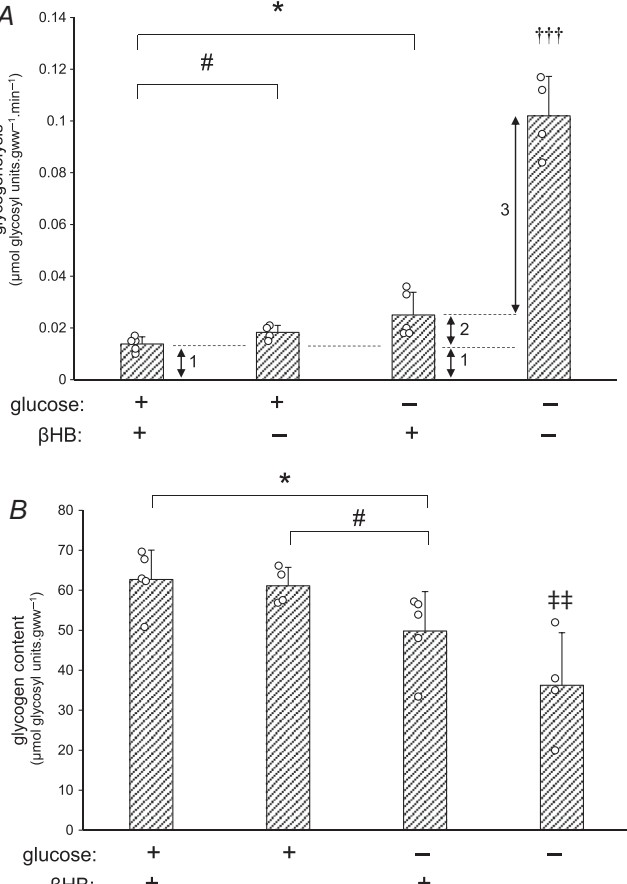

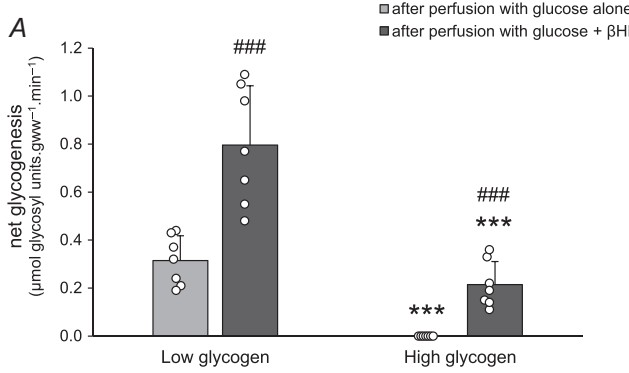

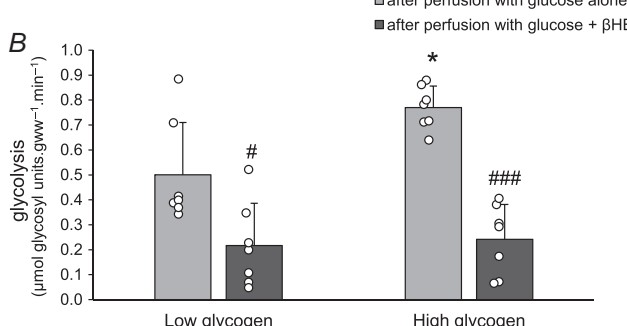

**Figure 4. Net glycogenesis and glycolysis in isolated perfused rat hearts**

*A*, net glycogenesis in low and high glycogen hearts after perfusion with glucose alone or glucose and $\beta$HB. Glycogen was estimated from total glycogen content before and after the perfusion; for further details see text. Data analysis by ANOVA and Tukey correction: *P*: LG Gluc *vs*. HG Gluc: <0.0001; LG $\beta$HB *vs*. HG $\beta$HB: <0.0001; LG Gluc *vs*. LG $\beta$HB: 0.0005; HG Gluc *vs*. HG $\beta$HB: <0.0001. *B*, glycolysis in low and high glycogen hearts was measured by perfusing hearts with [3H]glucose and quantifying 3H$_2$O production. For further details see text. Data analysis by ANOVA and Tukey correction: *P*: LG Gluc *vs*. HG Gluc: 0.019; LG $\beta$HB *vs*. HG $\beta$HB: 0.991; LG Gluc *vs*. LG $\beta$HB: 0.013; HG Gluc *vs*. HG $\beta$HB: <0.0001. Values are means ±SD. Effect of glycogen content is indicated as: *$P < 0.05$, **$P < 0.01$. Effect of $\beta$HB is indicated as: #$P < 0.05$, ###$P < 0.001$. Significance was taken at $P < 0.05$. $\beta$HB, after perfusion with glucose + $\beta$HB; Gluc, after perfusion with glucose alone; HG, high glycogen; LG, low glycogen.

**Figure 5. Glycogenolysis and glycogen content in hearts following 3H-labelling of glycogen**

Rates of glycogenolysis (*A*) and glycogen content (*B*) were measured in hearts pre-loaded with labelled glycogen (high glycogen: HG), which were subsequently perfused with buffer containing 11 mM glucose plus 4 mM $\beta$HB, 11 mM glucose alone, 4 mM $\beta$HB alone, or substrate-free buffer, during the subsequent chase. 1: basal glycogenolysis in the presence of ample exogenous substrate, corresponding to basal anaplerosis (Fig. 2: $f'_B$); 2: additional glycogenolysis in presence of cataplerotic substrate only, corresponding to anaplerosis required for $\beta$HB metabolism (Fig. 2: $f'_{KB}$); 3: additional glycogenolysis representing glycogen provision for oxidative metabolism (Fig. 2: *e'*). For further details see text. Values are means ± SD. *A*, data analysis by ANOVA and Tukey correction: *P*: +Gluc +$\beta$HB *vs*. +Gluc: 0.047; +Gluc +$\beta$HB *vs*. + $\beta$HB: 0.026; +Gluc +$\beta$HB *vs*. no substrate: <0.0001; +Gluc *vs*. +$\beta$HB: 0.187; +Gluc *vs*. no substrate: <0.0001; +$\beta$HB *vs*. no substrate: <0.0001. *B*, data analysis by ANOVA and Tukey correction: *P*: +Gluc +$\beta$HB *vs*. +Gluc: 0.720; +Gluc +$\beta$HB *vs*. + $\beta$HB: 0.047; +Gluc +$\beta$HB *vs*. no substrate: 0.009; +Gluc *vs*. +$\beta$HB: 0.037; +Gluc *vs*. no substrate: 0.009; +$\beta$HB *vs*. no substrate: 0.151. Effect of glucose is indicated as: *$P < 0.05$. Effect of $\beta$HB is indicated as: #$P < 0.05$. Difference between substrate-free perfusion (−Gluc −$\beta$HB) and substrate-containing perfusions is indicated as: †††$P < 0.001$. Difference between substrate-free perfusion (−Gluc −$\beta$HB) and glucose-containing perfusions is indicated as: ‡‡$P < 0.01$. Significance was taken at $P < 0.05$. $\beta$HB, D-$\beta$-hydroxybutyrate.

cytosolic free [NAD$^+$]/[NADH], calculated from the [lactate]/[pyruvate] ratio (Sato et al., 1995) (Fig. 6; Veech et al., 1969; Williamson et al., 1967).

In hearts perfused with $\beta$HB alone ($-$Gluc $+\beta$HB; a condition requiring increased anaplerosis) glycogen was mobilised (Figs 5 and 6), and glucose 1-phosphate

and glucose 6-phosphate were raised, but fructose 1,6-bisphosphate, 3-phosphoglycerate and lactate were all low; consequently the cytosolic free [NAD$^+$]/[NADH] ratio was reduced (less oxidised; Fig. 6). As expected, $\beta$HB levels measured in this metabolomic analysis were high, confirming the efficacy of this technique. Thus,

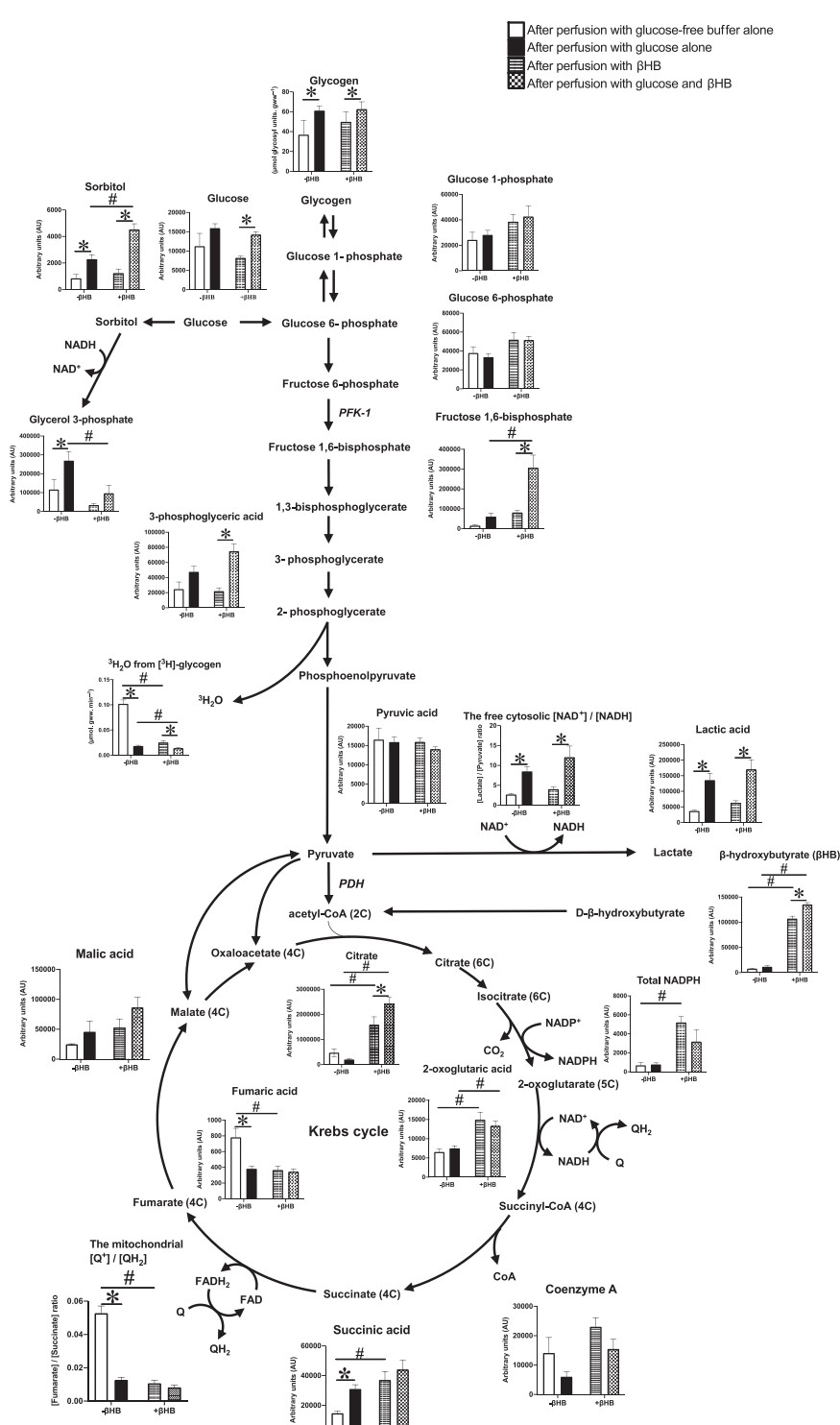

**Figure 6. Glycolytic and Krebs cycle intermediates in glycogen replete hearts after measurement of glycogenolysis**

Hearts were perfused with buffer containing glucose-free buffer alone ($-$Gluc $-\beta$HB), 11 mM glucose alone ($+$Gluc $-\beta$HB), 4 mM $\beta$HB alone ($-$Gluc $+\beta$HB), or 11 mM glucose plus 4 mM ($+$Gluc $+\beta$HB) over 40-min ($n$ = 4–5/group). For details of substrate analysis, see text. Values are means ± SD. Data analysis was by ANOVA with Tukey correction. Effect of glucose is indicated as: *$P$ < 0.05. Effect of $\beta$HB is indicated as: #$P$ < 0.05. Significance was taken at $P$ < 0.05. $\beta$HB, D-$\beta$-hydroxybutyrate; GDP, guanosine diphosphate; PFK-1, phosphofructokinase-1; PGK, 3-phosphoglycerate kinase; Q, ubiquinone; QH$_2$, ubiquinol; FAD(H), flavin adenine dinucleotide (reduced).

$\beta$HB oxidation was associated with increased citrate, 2-oxoglutarate and the total NADP (NADP/H) pool. $\beta$HB oxidation was associated with decreased fumarate, which in turn was associated with increased succinate. Thus, $\beta$HB oxidation decreased the mitochondrial free $[Q^+]/[QH_2]$ ratio (Fig. 6). In hearts perfused with both $\beta$HB and glucose (+Gluc +$\beta$HB), glucose, sorbitol, glucose 1-phosphate, glucose 6-phosphate, fructose 1,6-bisphosphate, 3-phosphoglycerate and lactate levels were high, and the cytosolic free $[NAD^+]/[NADH]$ ratio was increased (more oxidised). $\beta$HB oxidation was associated with increased citrate and 2-oxoglutarate levels, and a low mitochondrial $[Q^+]/[QH_2]$ ratio (Fig. 6).

### The effect of $\beta$HB oxidation on myocardial energetics using $^{31}$P-NMR spectroscopy

Figure 7 shows examples of $^{31}$P-NMR spectra for each perfusion condition: high and low glycogen content, perfused with glucose with or without $\beta$HB. The integrated areas of the ATP ($\gamma$-ATP, $\beta$-ATP and $\alpha$-ATP) resonance peaks was the same in all spectra, and the cytosolic pH, estimated for the $P_i$ resonance shifts, was similar in all hearts. As expected, high glycogen content was associated with increased phosphocreatine content

and phosphorylation potential ($[ATP]/[ADP][P_i]$), and also the free energy of ATP hydrolysis, $\Delta G_{ATP}$, together with decreased $P_i$, ADP and AMP. Perfusion with $\beta$HB significantly increased the myocardial PCr and decreased the free ADP and AMP concentrations, but only in high glycogen content hearts; $\beta$HB had no significant effect on the $^{31}$P-NMR spectra of hearts with depleted glycogen (Fig. 8). In hearts with high glycogen content, $\beta$HB also markedly increased the cytosolic phosphorylation potential ($[ATP]/[ADP][P_i]$) and the free energy of ATP hydrolysis, $\Delta G_{ATP}$ (Fig. 8).

Thus, high endogenous glycogen content was associated with increased myocardial $\beta$HB oxidation rates (Fig. 5), Krebs cycle intermediates (Fig. 6) and free energy of ATP hydrolysis $\Delta G_{ATP}$ (Fig. 8).

## Discussion

Recently there has been considerable interest in the potential role of $\beta$HB as an energetic substrate in the heart, particularly in the context of heart dysfunction, where substrate selection and oxidation may have a critical impact on cardiac function. Facilitating substrate utilisation, for example $\beta$HB oxidation, offers the possibility of optimising cardiac performance and treating

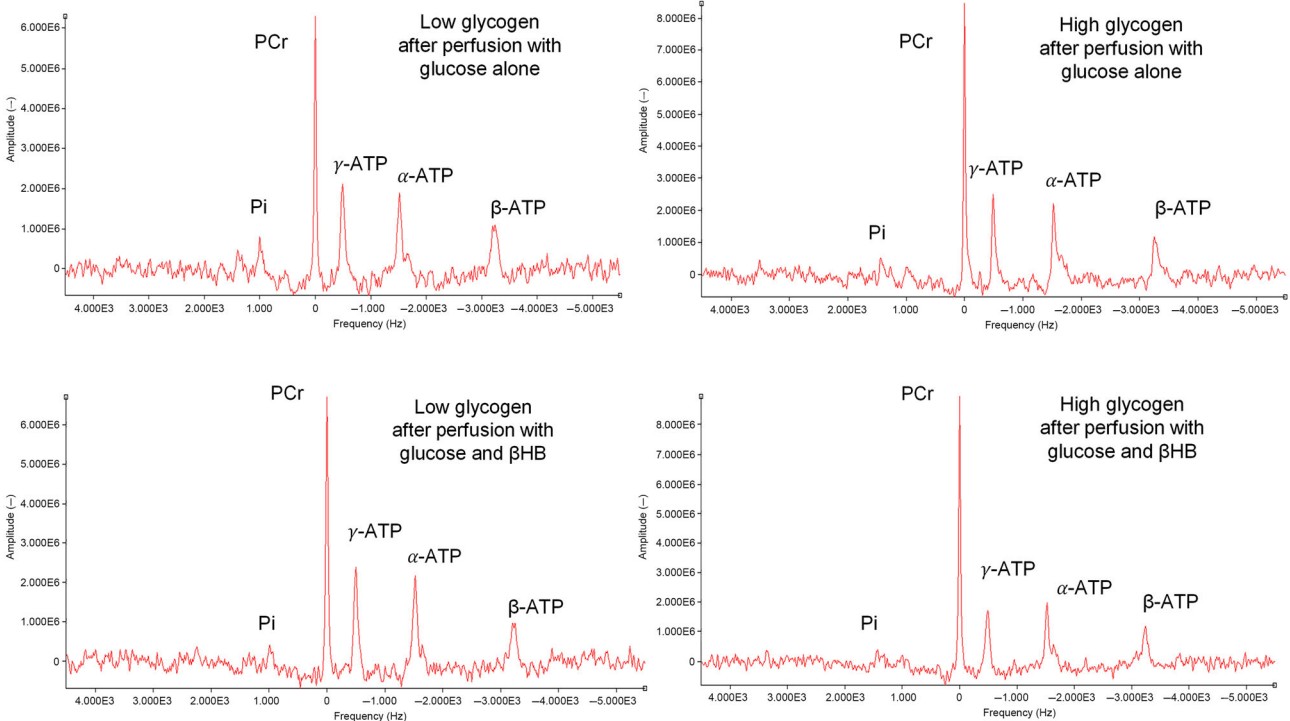

**Figure 7. Examples of $^{31}$P-NMR spectra acquired in hearts with high or low glycogen perfused with either 11 mM glucose alone or 11 mM glucose plus 4 mM $\beta$HB (*n* = 10 per group; see text for further details)**
The phosphocreatine (PCr) resonance frequency was set at 0 Hz. $\beta$HB, D-$\beta$-hydroxybutyrate. [Colour figure can be viewed at wileyonlinelibrary.com]

heart failure. Early pioneering work by Taegmeyer and colleagues (Russell & Taegtmeyer, 1991a, 1991b, 1992; Russell et al., 1997; Taegtmeyer, 1983; Taegtmeyer et al., 1980) indicated that carbohydrate provision, and potentially tissue glycogen, were important in acetoacetate metabolism, and hence of ketone bodies generally. The main findings of the present study, which was designed to examine specifically the metabolic role of myocardial glycogen in *β*HB metabolism and oxidation, and glycolytic rates (and hence the contribution of endogenous glycogen to cardiac ketone body metabolism), were that *β*HB oxidation is enhanced in hearts with higher glycogen content, and that *β*HB oxidation decreases glycolysis in hearts, redirecting exogenous glucose into

glycogen synthesis. Furthermore, we demonstrate for the first time that *β*HB provision leads to augmentation of glycogenolysis–anaplerosis, supporting our hypothesis. Hence myocardial glycogen facilitates cardiac *β*HB oxidation, which in turn facilitates Glycogen mobilisation for anaplerosis (for schematic overview see Graphical Abstract). We proposed that myocardial glycogen is an important anaplerotic substrate, via glycogenolysis and glycolysis to pyruvate, and thence to oxaloacetate, in order to maintain metabolic efficiency and support the heart's ability to oxidise (cataplerotic) *β*HB. Therefore, a novel pulse–chase technique was devised in order to measure anaplerosis indirectly from [3H]glycogen in glycogen-replete hearts perfused with *β*HB, and

**Figure 8. Effects of D-*β*-hydroxybutyrate (*β*HB) oxidation on energetic parameters in low and high glycogen hearts measured using ³¹P-NMR spectroscopy**

*A*, ATP. *B*, phosphocreatine (PCr). *C*, inorganic phosphate (P$_i$). *D*, cytosolic pH. *E*, ADP. *F*, AMP. *G*, phosphorylation potential (estimated as [ATP]/[P$_i$].[ADP]). *H*, free energy of ATP hydrolysis ($\Delta G_{ATP}$). Data are means ± SD. Data analysis by ANOVA and Tukey correction: *A*, *P*: LG +Gluc *vs*. LG +*β*HB: 0.866; LG +Gluc *vs*. HG +Gluc: 0.810; LG +*β*HB *vs*. HG +*β*HB: 0.941; HG +Gluc *vs*. HG +*β*HB: 0.883. *B*, *P*: LG +Gluc *vs*. LG +*β*HB: 0.818; LG +Gluc *vs*. HG +Gluc: 0.784; LG +*β*HB *vs*. HG +*β*HB: <0.0001; HG +Gluc *vs*. HG +*β*HB: 0.016. *C*, *P*: LG +Gluc *vs*. LG +*β*HB: 0.161; LG +Gluc *vs*. HG +Gluc: 0.193; LG +*β*HB *vs*. HG +*β*HB: 0.022; HG +Gluc *vs*. HG +*β*HB: 0.789. *D*, *P*: LG +Gluc *vs*. LG +*β*HB: 0.999; LG +Gluc *vs*. HG +Gluc: 0.250; LG +*β*HB *vs*. HG +*β*HB: 0.553; HG +Gluc *vs*. HG +*β*HB: 0.907. *E*, *P*: LG +Gluc *vs*. LG +*β*HB: 0.201; LG +Gluc *vs*. HG +Gluc: 0.249; LG +*β*HB *vs*. HG +*β*HB: 0.021; HG +Gluc *vs*. HG +*β*HB: 0.009. *F*, *P*: LG +Gluc *vs*. LG +*β*HB: 0.168; LG +Gluc *vs*. HG +Gluc: 0.189; LG +*β*HB *vs*. HG +*β*HB: 0.017; HG +Gluc *vs*. HG +*β*HB: 0.003. *G*, *P*: LG +Gluc *vs*. LG +*β*HB: 0.609; LG +Gluc *vs*. HG +Gluc: 0.049; LG +*β*HB *vs*. HG +*β*HB: 0.011; HG +Gluc *vs*. HG +*β*HB: 0. 051. *H*, *P*: LG +Gluc *vs*. LG +*β*HB: 0.889; LG +Gluc *vs*. HG +Gluc: 0.776; LG +*β*HB *vs*. HG +*β*HB: 0.012; HG +Gluc *vs*. HG +*β*HB: 0.038. The effect of glycogen content is indicated as: *$P < 0.05$, ***$P < 0.001$; the effect of *β*HB is indicated as: #$P < 0.05$, ##$P < 0.01$. +*β*HB, perfusion with glucose + *β*HB; +Gluc, perfusion with glucose alone; LG, low glycogen; HG, high glycogen.

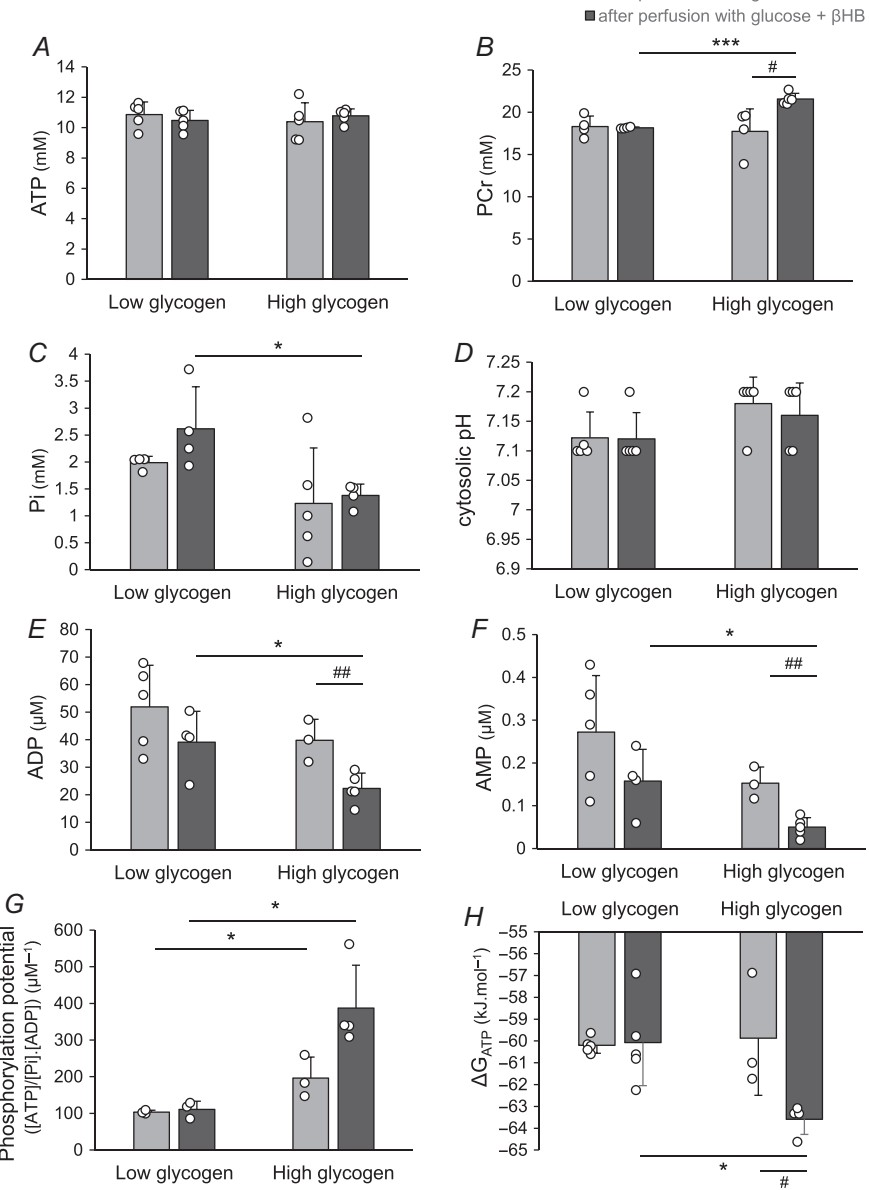

is reported here. We show here that glycogenolysis from HG hearts with [³H]glycogen perfused with $\beta$HB alone was two-fold higher than similar hearts perfused with glucose plus $\beta$HB. Metabolomic analysis indicated that the increase in glycogenolysis significantly increased the utilisation of $\beta$HB, via increased fructose 1,6-bisphosphate and 3-phosphoglycerate. Furthermore, $\beta$HB oxidation was associated with increased Krebs cycle intermediates, citrate, 2-oxoglutarate and succinate, together with total NADP(H). $\beta$HB oxidation decreased the mitochondrial free $[Q^+]/[QH_2]$ ratio, thereby increasing the redox potential of the mitochondrial Q couple, as proposed by Sato and co-workers (Sato et al., 1995). Ultimately, $\beta$HB increased the free energy of ATP hydrolysis ($\Delta G_{ATP}$) by increasing [PCr] and the phosphorylation potential in high glycogen hearts, but not in low glycogen hearts, supporting our proposals for the role of glycogen in ketone body metabolism.

Perfusion of HG hearts with glucose alone did not further increase myocardial glycogen, likely due to the upper limit of glycogen content having been attained (Cross et al., 1996). However, perfusion with $\beta$HB in addition to the glucose significantly increased glycogen in LG hearts (Fig. 3), suggesting that $\beta$HB oxidation facilitated increased glycogen synthesis in glycogen-depleted myocardium by decreasing glycolysis through inhibition of PDH (Murray et al., 2016; Robinson & Williamson, 1980), glyceraldehyde-3-phosphate dehydrogenase (GAPDH) (Mochizuki & Neely, 1979) or phosphofructokinase (PFK) (Murray et al., 2016; Robinson & Williamson, 1980), which was associated with increased cytosolic free $[NAD^+]/[NADH]$ and lactate production (Laughlin et al., 1994) – $\beta$HB is both an energetic substrate and is glucose-sparing for energy provision. It is postulated that $\beta$HB metabolism inhibits the activity of PDH, via activation of PDH kinase activity (Randle et al., 1964; Robinson & Williamson, 1980). Such metabolic effects of $\beta$HB on glycogen synthesis were also found in dog and perfused working rat hearts (Kashiwaya et al., 1994; Laughlin et al., 1994).

Glycolysis from exogenous glucose in LG hearts perfused with glucose alone was significantly lower than in HG hearts perfused with glucose because some of the exogenous glucose assimilated was preferentially used to replenish glycogen. This suggests that glucose utilisation from exogenous [5-³H]glucose must be interpreted as the sum of net glycogenesis and glycolytic rates (assuming no other major metabolic fates of exogenous glucose, e.g. the pentose phosphate pathway, which accounts for the disposal of some exogenous glucose, but was not measured in the present study due to its known relatively low activity in heart; Zimmer, 1992). The lower glycolysis in LG hearts perfused with $\beta$HB resulted from some exogenous glucose being used to replenish myocardial glycogen following the preceding glycogen depletion regime, as indicated by the

increase in net glycogenesis rate from 0.3 to 0.8 $\mu$mol glycosyl units gww$^{-1}$ min$^{-1}$.

As expected, addition of $\beta$HB decreased glycolysis in all hearts, as previous work has shown that $\beta$HB inhibits the activity of multiple glycolytic enzymes, including PDH (Murray et al., 2016; Robinson & Williamson 1980), GAPDH (Mochizuki & Neely 1979) and PFK (Murray et al., 2016; Robinson & Williamson 1980). $\beta$HB oxidation rates in high glycogen hearts with high anaplerotic potential were two-fold higher than in low glycogen hearts. High $\beta$HB oxidation rates significantly decreased exogenous glucose utilisation (0.44 *vs.* 1.0 $\mu$mol glycosyl units gww$^{-1}$ min$^{-1}$), potentially due to inhibition of glucose uptake (Randle et al., 1964; Robinson & Williamson 1980). Hence high glycogen content augmented $\beta$HB oxidation, via glycogenolysis and subsequent glycolysis to pyruvate and anaplerosis to oxaloacetate. We reasoned that the principal metabolic fate of endogenous glycogen is glycolysis, then anaplerosis, with or without oxidation depending on other (carbohydrate) substrate availability (see Fig. 2). Therefore, we estimated anaplerosis from glycogen in hearts oxidising ketone body by measuring glycogenolysis from pre-labelled glycogen in hearts perfused with $\beta$HB. Glycogenolysis in HG hearts perfused with $\beta$HB alone (without exogenous glucose as an energetic source) was two-fold higher than in hearts perfused with glucose and $\beta$HB, thereby decreasing myocardial glycogen by about a quarter over the course of the perfusion (Fig. 5). The net rate of glycogenolysis, and hence potential anaplerotic flux from [³H]glycogen, was 0.026 $\pm$ 0.004 $\mu$mol glycosyl units gww$^{-1}$ min$^{-1}$ and hence quantitatively significant in terms of depot provision and residual glycogen depletion. Moreover, glycogenolysis in HG hearts perfused with glucose-free buffer was significantly higher than in those hearts perfused with glucose. The labelled glycogen was preferentially oxidised in the absence of substrates, and was associated with rapidly decreasing glycogen content (Goodwin et al., 1996). Taken together, these results indicate that myocardial glycogen provides both anaplerotic substrate and energetic (oxidisable) substrate, and that its disposition is affected by both exogenous glucose and $\beta$HB availability.

An attempt was made to assess metabolic flux in glycogen-replete hearts perfused with and without $\beta$HB through the key pathways of glycolysis and the TCA cycle using targeted metabolomic analysis. $\beta$HB oxidation increased the Krebs cycle intermediates citrate, 2-oxoglutarate and succinate, and also the total NADP(H) pool. These increased Krebs cycle intermediates in $\beta$HB-perfused hearts indicate a mismatch between synthesis and removal, favouring accumulation and reflecting differences in enzyme activities, and can be cautiously interpreted as increased TCA flux from $\beta$HB substrate; however, ketone body metabolism also requires

CoASH from succinyl-CoA (emphasising the close inter-relationship between ketone body oxidation and Krebs cycle function) and this can have implications for TCA cycle intermediate levels. $\beta$HB oxidation increased the production of succinate from succinyl-CoA coupled with substrate-level phosphorylation of guanosine diphosphate (GDP) to GTP catalysed by succinyl-CoA synthase (Kaufman, 1955; Kaufman et al., 1953). Moreover, succinyl-CoA synthase can replenish succinyl-CoA for $\beta$HB metabolism, whereby succinyl-CoA acts as a CoASH donor for esterification of $\beta$HB-derived acetoacetate: CoASH is transferred from succinyl-CoA to form acetoacetyl-CoA in the presence of 3-ketoacyl-CoA transferase (Ottaway et al., 1981). The increase in succinate was associated with decreased fumarate catalysed by the near-equilibrium succinate dehydrogenase reaction (Sato et al., 1995; Veech et al., 1969; Williamson et al., 1967). Thus, $\beta$HB oxidation decreased the mitochondrial free $[Q^+]/[QH_2]$ ratio (Pawlosky et al., 2017) (Fig. 6; Sato et al., 1995), which increased the free energy of ATP hydrolysis by increasing [PCr] and the phosphorylation potential (Fig. 8) in high glycogen content hearts. This was due to the reaction of mitochondrial creatine kinase, which catalyses the transfer of a phosphoryl group from ATP to form PCr. PCr is a transport molecule with the reverse reaction catalysed by myofibrillar creatine kinase to liberate ATP, which increases the cytosolic phosphorylation potential and the free energy of ATP hydrolysis (Sato et al., 1995), with potential impact on cardiac energetics, and ultimately cardiac mechanical function.

During the fed state, NADPH is primarily produced from glucose metabolism by the pentose phosphate pathway (Veech et al., 2017). However, NADPH can also be produced from metabolism of $\beta$HB in the Krebs cycle (Kashiwaya et al., 1997; Sato et al., 1995). During ketosis, mitochondrial sirtuin-3 (SIRT3) deacetylates and activates the NADP-dependent isocitrate dehydrogenase isoenzyme IDH2 (Someya et al., 2010), which increases 2-oxoglutarate ($\alpha$-ketoglutarate) and succinate measured using metabolomics. $\beta$HB inhibits class I and IIa histone deacetylases (HDACs), which increase the genes encoding oxidative stress resistance factors, forkhead box O3a (FOXO3a) and MT2 (Shimazu et al., 2013); FOXO3a induces the expression of a cytoplasmic form of NADP-dependent isocitrate dehydrogenase IDH1 (Charitou et al., 2015). The citrate or isocitrate from $\beta$HB metabolism in mitochondria can be exported by the citrate–isocitrate carrier to the cytosol for the production of NADPH by IDH1. This leads to increased NADPH production and an increased ratio of reduced to oxidised glutathione in mitochondria (Someya et al., 2010). NADPH provides the electrons for the reduction of molecular antioxidants (glutathione disulphide, glutathione, vitamins C and E), which decreases reactive oxygen or nitrogen species through glutathione peroxidase (Veech et al., 2017), and hence provides myocardial protection in ischaemia–reperfusion injury, and potentially also in heart failure.

Interestingly, perfusion with exogenous glucose significantly increased myocardial sorbitol, as well as lactate. Aldose reductase catalyses the NADP-dependent reduction of glucose to sorbitol (polyol pathway); subsequently, sorbitol can be metabolised to fructose and glycerol 3-phosphate in NAD-dependent cytosolic reactions. We show here that these hearts also have raised glycerol 3-phosphate, which was associated with the increased cytosolic free $[NAD^+]/[NADH]$ ratio (Tran & Wang, 2019). Glycerol 3-phosphate is required for esterification of fatty acids, the main energetic substrate of heart (and also, like ketone bodies, cataplerotic), and heart does not express glycerol kinase to a significant extent, hence this is likely to be an important myocardial pathway (Tran & Wang, 2019).

In the absence of $\beta$HB, glucose alone successfully maintained the free energy of ATP hydrolysis ($\Delta G_{ATP}$) in both low and high glycogen hearts by maintaining [PCr] and phosphorylation potential, as well as cardiac mechanical function. Hence, these data demonstrate that despite changing glycogen content, the metabolic integrity of the myocardium was maintained intact under baseline conditions; however, $\beta$HB utilisation increased free energy of ATP hydrolysis between low and high glycogen hearts by increasing [PCr] and phosphorylation potential. Thus, glycogen and glucose are both important anaplerotic substrates for oxidation of $\beta$HB in the heart. However, $\beta$HB was unable to increase the free energy of ATP hydrolysis in low glycogen hearts (Fig. 8) due to the low glycolytic flux and anaplerotic potential (Fig. 4*B* and Table 2).

One caveat of this study is that the perfusion buffer did not contain fatty acid, a major, but cataplerotic, fuel of the heart (Stanley et al., 2005). In physiological conditions *in vivo*, glucose, $\beta$HB and fatty acids are all available in the plasma; under starvation conditions, non-esterified fatty acid and KB concentrations both increase, as does myocardial glycogen content. Fatty acids and ketone bodies are both 2-carbon unit equivalents and, had they been used together, they would have introduced complex cataplerotic substrate competition, making the interpretation of results difficult. Furthermore, we have assumed that the only metabolic fates of mobilised endogenous cardiac glycogen are oxidation and anaplerosis, via glycolysis (myocardium having low flux rates of pentose phosphate pathway, and is not lipogenic); lactate production was related to glucose status (Fig. 6) and was also discounted in this analysis.

Recently, the putative role of $\beta$-hydroxybutyrate as a myocardial oxidative substrate has been emphasised in studies on the positive role of ketone bodies in supporting

cardiac function in heart failure (Aubert et al., 2016; Bedi et al., 2016); the mechanistic linkage between cardiac glycogen and myocardial ketone oxidation demonstrated here offers the possibility for therapeutic intervention at the level of cardiac carbohydrate storage in order to potentially maximise the accessibility of this important substrate in health and disease.

In conclusion, using a novel approach to estimate anaplerosis from tissue glycogen, glycolytic flux from myocardial glycogen increased the heart's ability to oxidise $\beta$HB, and $\beta$HB oxidation increased the mitochondrial redox potential, ultimately increasing the free energy of ATP hydrolysis (see Abstract Figure).

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

## Additional information

### Data availability statement

The data that support the findings of this study are available on request from the corresponding author (rhys.evans@dpag.ox.ac.uk), as well as the Statistical Summary Document.

### Competing interests

The authors declare that there are no commercial or financial relationships that could be construed as a potential conflict of interest.

### Author contributions

Conceptualisation: A.A.K, B.J.S., R.D.E., K.C. Methodology: A.A.K, B.J.S., C-R.C, C.C., M.C., D.H., J.M, R.D.E. Investigation: A.A.K, B.J.S., C-R.C, H.L., C.C., D.H., R.D.E. Formal analysis: A.A.K, B.J.S., M.C., C.C., D.H., J.M, R.D.E. Writing – original draft: R.D.E., K.C. Writing – review and editing: A.A.K, B.J.S., C-R.C, H.L., M.C., C.C., D.H., J.M., R.D.E., K.C. Project administration: J.M, R.D.E., K.C. Funding acquisition: A.A.K., B.J.S., J.M., R.D.E., K.C. Supervision: J.M., R.D.E., K.C. All authors have read, revised and approved the final version of this manuscript and agree to be accountable for all aspects of the work in ensuring that questions related to the accuracy or integrity of any part of the work are appropriately investigated and resolved. All persons designated as authors qualify for authorship, and all those who qualify for authorship are listed.

### Funding

This work was supported by a British Heart Foundation Project Grant PG/07/030/22667 (R.D.E). The authors are grateful for the support of the BHF to permit these studies to be performed.

### Acknowledgements

A.A.K. thanks Malaysian Government for a Ministry of Higher Education Scholarship (MyBrainSc) for his DPhil studentship. B.J.S. thanks the Royal Commission for the Exhibition of 1851 for her DPhil studentship. We are indebted to Pete Cox for helpful discussions during the conceptualisation of this study.

### Keywords

D-$\beta$-hydroxybutyrate oxidation, glycogen, glycolysis, metabolomics, redox states

## Supporting information

Additional supporting information can be found online in the Supporting Information section at the end of the HTML view of the article. Supporting information files available:

**Statistical Summary Document**
**Peer Review History**

