## [Peer Review History · The Journal of Physiology]

On the interdependence of ketone body oxidation, glycogen content, glycolysis and energy metabolism in the heart

Rhys David Evans, Azrul Abdul Kadir, Brianna Stubbs, Cher-Rin Chong, Henry Lee, Mark Cole, Carolyn A Carr, David Hauton, James McCullagh, and Kieran Clarke

DOI: 10.1113/JP284270

Corresponding author(s): Azrul Kadir (azrulabdulkadir@gmail.com)

The following individual(s) involved in review of this submission have agreed to reveal their identity: Kyle S McCommis (Referee #1)

Review Timeline:

Submission Date:	23-Oct-2022
Editorial Decision:	06-Nov-2022
Resubmission Received:	14-Dec-2022
Editorial Decision:	18-Jan-2023
Revision Received:	20-Jan-2023
Accepted:	23-Jan-2023

Senior Editor: Michael Hogan

Reviewing Editor: Bettina Mittendorfer

Transaction Report:

Dear Dr Evans,

Re: JP-RP-2022-284017 "On the interdependence of ketone body oxidation, glycogen content, glycolysis and energy metabolism in the heart" by Rhys David Evans, Azrul Abdul Kadir, Brianna Stubbs, Cher-Rin Chong, Henry Lee, Mark Cole, Carolyn A Carr, David Hauton, James McCullagh, and Kieran Clarke

Thank you for submitting your manuscript to The Journal of Physiology. It has been assessed by a Reviewing Editor and by 2 Referees and the reports are copied below.

Please let your co-authors know of the following editorial decision as quickly as possible.

As you will see, in its current form, the manuscript is not acceptable for publication in The Journal of Physiology. In comments to me, the Reviewing Editor expressed interest in the potential of this study, but much work still needs to be done (and this may include new experiments) in order to satisfactorily address the concerns raised in the reports.

In view of this interest, I would like to offer you the opportunity to carry out all of the changes requested in full, and to resubmit a new manuscript using the "Submit Special Case Resubmission for JP-RP-2022-284017..." on your homepage.

We cannot, of course, guarantee ultimate acceptance at this stage as the revisions required are substantial. However, we encourage you to consider the requested changes and resubmit your work to us if you are able to complete or address all changes.

A new manuscript would be renumbered and redated, but the original referees would be consulted wherever possible. An additional referee's opinion could be sought, if the Reviewing Editor felt it necessary. A full response to each of the reports should be uploaded with a new version.

I hope that the points raised in the reports will be helpful to you.

Yours sincerely,

Michael C. Hogan
Senior Editor
The Journal of Physiology
<https://jp.msubmit.net>
<http://jp.physoc.org>
The Physiological Society
Hodgkin Huxley House
30 Farringdon Lane
London, EC1R 3AW
UK
<http://www.physoc.org>
<http://journals.physoc.org>

EDITOR COMMENTS

Reviewing Editor:

The reviewers have provided constructive feedback that will help the authors further improve an already strong paper

The authors are encouraged to address the perceived lack of novelty in their response to the reviewers and Editors and also the revised paper

Senior Editor:

Please comply with the Statistics Policy.

Comments to the Author:

While both reviewers and editors feel that the study has merit, there are significant concerns as to whether the study provides sufficient new novelty to merit publication in the Journal of Physiology. With this in mind, we would consider a resubmission of your manuscript if you feel you can address this and the other concerns as outlined in the reviews.

REFEREE COMMENTS

Referee #1:

The authors postulate that glycogen breakdown and pyruvate anaplerosis would enhance the hearts ability to oxidize the ketone body beta-hydroxybutyrate, and conversely that glucose/ketone metabolism would maintain cardiac glycogen levels. This is an interesting study that appears well performed, however it should be noted that these results are largely confirmatory of work performed predominantly in the Hans Krebs and then Heinrich Taegtmeyer labs in the 1980s-1990s, and indeed, some of this work is cited in this manuscript. This historic literature starts with the finding that isolated hearts cannot maintain work when perfused with ketone bodies alone (Taegtmeyer, Hems, and Krebs, *Biochem J* 1980;186:701, and reference 7). However, the hearts being perfused with ketones are rescued if also perfused with glucose or pyruvate, due to pyruvate carboxylation and anaplerosis (references above and references 8 and 9). Lastly, many of the connections to glycogen metabolism/storage after glucose and/or ketone infusion were also identified (addition of ketone bodies to glucose perfusate increases glycogen synthesis and decreases glycogenolysis and glycolysis: references 36 and 42; and glucose incorporation into glycogen and increased glycogen content if hearts also infused with acetoacetate: Russell RR, et al. *JCI* 1997;100:2892). Some specific suggestions for revising this work:

- 1) The protocol to increase or decrease cardiac glycogen levels involving substrate-free perfusion vs glucose, insulin, pyruvate, and lactate somewhat complicates some of these analyses as surely increased glycogen storage is not the only effect of prolonged exposure to no vs surplus metabolic substrates (for example, different nucleotide redox ratios?). Perhaps another way to perform a similar analysis would be to have all hearts with normal/similar glycogen content, but then infuse hearts with a glycogen phosphorylase inhibitor to prevent glycogenolysis.
- 2) In the perfusate to create high glycogen, it seems strange that pyruvate/lactate is provided at a 4:1 ratio. Typically lactate/pyruvate is provided at nearly 10:1 ratio to better mimic in vivo concentrations and better maintain redox ratios.
- 3) Perhaps they are also not different, but it would be beneficial to display both the heart rates and developed pressures in addition to RPP in Table 1.
- 4) The figure legends appear to provide the total number of animals used in the experiment and not the n per each group, which needs to be corrected.
- 5) The quantified glycogen concentrations appear very high for the heart in my experience, even for the "low glycogen" group. For mouse hearts, glycogen is typically under 10 $\mu\text{mol glycosyl/g}$ wet weight, but perhaps there is a large difference between mouse and rat heart glycogen that I am not aware of. Conversely, is it possible that these units are actually per gram dry weight instead of wet weight? It would also be somewhat interesting to see "normal" rat heart glycogen concentrations here without any infusions altering glycogen abundance.
- 6) Along the lines with the last comment above, there are some instances where statements are not technically correct. On page 11: "Initial perfusion with glucose-free buffer decreased myocardial glycogen"... this phrasing is not correct since it is not a pre vs post measurement but is instead comparing the glycogen content after no substrate perfusion vs high substrate perfusion. These types of statements are also made on page 14: "perfusion of high glycogen hearts with substrate-free buffer increased fumarate, which associated with decreased succinate". -Yes, those trends are true for comparing to perfusions with glucose and/or bHB, but the statement seems more of a comparison to what the heart content is before perfusing with substrate-free buffer. The wording just needs to be corrected here to compare the different substrate perfusions.

Referee #2:

A. Kadir and colleagues have done a study in rat hearts investigating the importance of glycogen presence for substrate metabolism with special regards to glucose and ketone bodies.

The study is very well designed to answer the research question. I have only a few point.

Results:

Fig. 3C. In the legends it says N=42. To me it seems like only 10 hearts have been plottet? Could the authors explain?

It seems Pete Cox was a part of the study back in 2018, when it was presented as a poster: P31 Effect of glycogen content

on ketone body oxidation and glycolysis in the isolated rat heart. How comes he is not an author in the final version?

Could the authors maybe discuss the clinical impact of this finding in the paper. Should this be used a tool to decide whom would benefit from ketone bodies? Should glycogen build be attained?

ADDITIONAL FORMATTING REQUIREMENTS FOR RESUBMISSION:

-Include a Key Points list in the article itself, before the Abstract.

-Author photo and profile. First (or joint first) authors are asked to provide a short biography (no more than 100 words for one author or 150 words in total for joint first authors) and a portrait photograph. These should be uploaded and clearly labelled with the revised version of the manuscript. See Information for Authors for further details.

-You must start the Methods section with a paragraph headed Ethical Approval. A detailed explanation of journal policy and regulations on animal experimentation is given in Principles and standards for reporting animal experiments in The Journal of Physiology and Experimental Physiology by David Grundy J Physiol, 593: 2547-2549. doi:10.1113/JP270818.). A checklist outlining these requirements and detailing the information that must be provided in the paper can be found at: <https://physoc.onlinelibrary.wiley.com/hub/animal-experiments>. Authors should confirm in their Methods section that their experiments were carried out according to the guidelines laid down by their institution's animal welfare committee, and conform to the principles and regulations as described in the Editorial by Grundy (2015). The Methods section must contain details of the anaesthetic regime: anaesthetic used, dose and route of administration and method of killing the experimental animals.

-The Reference List must be in Journal format

-Your manuscript must include a complete Additional Information section

-Please upload separate high-quality figure files via the submission form.

-Please ensure that the Article File you upload is a Word file.

-A Statistical Summary Document, summarising the statistics presented in the manuscript, is required upon revision. It must be on the Journal's template, which can be downloaded from the link in the Statistical Summary Document section here: https://jp.msubmit.net/cgi-bin/main.plex?form_type=display_requirements#statistics

-Papers must comply with the Statistics Policy https://jp.msubmit.net/cgi-bin/main.plex?form_type=display_requirements#statistics

In summary:

-If n {less than or equal to} 30, all data points must be plotted in the figure in a way that reveals their range and distribution. A bar graph with data points overlaid, a box and whisker plot or a violin plot (preferably with data points included) are acceptable formats.

-If $n > 30$, then the entire raw dataset must be made available either as supporting information, or hosted on a not-for-profit repository e.g. FigShare, with access details provided in the manuscript.

- 'n' clearly defined (e.g. x cells from y slices in z animals) in the Methods. Authors should be mindful of pseudoreplication.

- All relevant 'n' values must be clearly stated in the main text, figures and tables, and the Statistical Summary Document (required upon revision)

- The most appropriate summary statistic (e.g. mean or median and standard deviation) must be used. Standard Error of the Mean (SEM) alone is not permitted.

- Exact p values must be stated. Authors must not use 'greater than' or 'less than'. Exact p values must be stated to three significant figures even when 'no statistical significance' is claimed.

- Statistics Summary Document completed appropriately upon revision

- A Data Availability Statement is required for all papers reporting original data. This must be in the Additional Information section of the manuscript itself. It must have the paragraph heading "Data Availability Statement". All data supporting the results in the paper must be either: in the paper itself; uploaded as Supporting Information for Online Publication; or archived in an appropriate public repository. The statement needs to describe the availability or the absence of shared data. Authors must include in their Statement: a link to the repository they have used, or a statement that it is available as Supporting Information; reference the data in the appropriate section(s) of their manuscript; and cite the data they have shared in the References section. Whenever possible the scripts and other artefacts used to generate the analyses presented in the paper should also be publicly archived. If sharing data compromises ethical standards or legal requirements then authors are not expected to share it, but must note this in their Statement. For more information, see our Statistics Policy.

- Please include an Abstract Figure file, as well as the figure legend text within the main article file. The Abstract Figure is a piece of artwork designed to give readers an immediate understanding of the research and should summarise the main conclusions. If possible, the image should be easily 'readable' from left to right or top to bottom. It should show the physiological relevance of the manuscript so readers can assess the importance and content of its findings. Abstract Figures should not merely recapitulate other figures in the manuscript. Please try to keep the diagram as simple as possible and without superfluous information that may distract from the main conclusion(s). Abstract Figures must be provided by authors no later than the revised manuscript stage and should be uploaded as a separate file during online submission labelled as File Type 'Abstract Figure'. Please ensure that you include the figure legend in the main article file. All Abstract Figures should be created using BioRender. Authors should use The Journal's premium BioRender account to export high-resolution images. Details on how to use and access the premium account are included as part of this email.

UNIVERSITY OF OXFORD

R.D. Evans
Reader
Telephone +44 (0) 1865 272445
Facsimile +44 (0) 1865 282272
rhys.evans@dpag.ox.ac.uk

Department of Physiology, Anatomy and Genetics
Sherrington Building
South Parks Road
Oxford U.K.
OX1 3PT

14th December 2022

Prof Michael Hogan
Senior Editor
The Journal of Physiology

Dear Prof Hogan

M/S/ JP-RP-2022-284017 “On the interdependence of ketone body oxidation, glycogen content, glycolysis and energy metabolism in the heart” Kadir et al.

Many thanks for allowing us the opportunity to re-submit this work in extensively revised form. We are grateful for the opportunity. We do so now, believing that we have answered all the (very reasonable and appropriate) queries raised by yourself and the Referees. We would be very grateful if the new M/S could be re-evaluated with a view to publishing in the Journal of Physiology. Obviously if there are further issues we will be very happy to try to address them but we think the M/S is much improved with the help of yourselves and the Referees. We are grateful for your time and help.

With many thanks and best wishes

Rhys

R.D. Evans BSc MB.BS MD D.Phil FRCA FFICM

Reader
Department of Physiology, Anatomy and Genetics
University of Oxford
Sherrington Building
South Parks Road
Oxford OX1 3PT

Kadir et al

On the interdependence of ketone body oxidation, glycogen content, glycolysis and energy metabolism in the heart

Original J Physiol submission: JP-RP-2022-284017

Authors responses to Referee's and Editor's comments:

Dear Dr Evans,

Re: JP-RP-2022-284017 "On the interdependence of ketone body oxidation, glycogen content, glycolysis and energy metabolism in the heart" by Rhys David Evans, Azrul Abdul Kadir, Brianna Stubbs, Cher-Rin Chong, Henry Lee, Mark Cole, Carolyn A Carr, David Hauton, James McCullagh, and Kieran Clarke

Thank you for submitting your manuscript to The Journal of Physiology. It has been assessed by a Reviewing Editor and by 2 Referees and the reports are copied below.

Please let your co-authors know of the following editorial decision as quickly as possible.

As you will see, in its current form, the manuscript is not acceptable for publication in The Journal of Physiology. In comments to me, the Reviewing Editor expressed interest in the potential of this study, but much work still needs to be done (and this may include new experiments) in order to satisfactorily address the concerns raised in the reports.

In view of this interest, I would like to offer you the opportunity to carry out all of the changes requested in full, and to resubmit a new manuscript using the "Submit Special Case Resubmission for JP-RP-2022-284017..." on your homepage.

We cannot, of course, guarantee ultimate acceptance at this stage as the revisions required are substantial. However, we encourage you to consider the requested changes and resubmit your work to us if you are able to complete or address all changes.

A new manuscript would be renumbered and redated, but the original referees would be consulted wherever possible. An additional referee's opinion could be sought, if the Reviewing Editor felt it necessary. A full response to each of the reports should be uploaded with a new version.

I hope that the points raised in the reports will be helpful to you.

Yours sincerely,

Michael C. Hogan
Senior Editor
The Journal of Physiology
<https://jp.msubmit.net>
<http://jp.physoc.org>
The Physiological Society
Hodgkin Huxley House
30 Farringdon Lane
London, EC1R 3AW
UK
<http://www.physoc.org>
<http://journals.physoc.org>

EDITOR COMMENTS

Reviewing Editor:

The reviewers have provided constructive feedback that will help the authors further improve an already strong paper

The authors are encouraged to address the perceived lack of novelty in their response to the reviewers and Editors and also the revised paper

Senior Editor:

Please comply with the Statistics Policy.

Comments to the Author:

While both reviewers and editors feel that the study has merit, there are significant concerns as to whether the study provides sufficient new novelty to merit publication in the Journal of Physiology. With this in mind, we would consider a resubmission of your manuscript if you feel you can address this and the other concerns as outlined in the reviews.

Thank you for your careful review of our manuscript. We are pleased that it has been found to have some merit and now resubmit a re-worked paper which we believe addresses both the Editor's concerns and answers all the Referee's comments, including the issue of novelty. We think it is much improved as a result and would be very grateful if it would be re-evaluated for publication in the Journal of Physiology.

Concerning novelty, we would kindly direct the Editor to our response to Referee #1 below, and the revised text.

REFEREE COMMENTS

We are extremely grateful for the Referee's thorough and insightful appraisal of the M/S, and valuable comments. We hope and believe we have addressed these, as detailed below under individual queries, and in the revised M/S text, and think as a result the M/S is considerably improved. Thank you.

Referee #1:

The authors postulate that glycogen breakdown and pyruvate anaplerosis would enhance the hearts ability to oxidize the ketone body beta-hydroxybutyrate, and conversely that glucose/ketone metabolism would maintain cardiac glycogen levels. This is an interesting study that appears well performed, however it should be noted that these results are largely confirmatory of work performed predominantly in the Hans Krebs and then Heinrich Taegtmeyer labs in the 1980s-1990s, and indeed, some of this work is cited in this manuscript. This historic literature starts with the finding that isolated hearts cannot maintain work when perfused with ketone bodies alone (Taegtmeyer, Hems, and Krebs, *Biochem J* 1980;186:701, and reference 7). However, the hearts being perfused with ketones are rescued if also perfused with glucose or pyruvate, due to pyruvate carboxylation and anaplerosis (references above and references 8 and 9). Lastly, many of the connections to glycogen metabolism/storage after glucose and/or ketone infusion were also identified (addition of ketone bodies to glucose perfusate increases glycogen synthesis and decreases glycogenolysis and glycolysis: references 36 and 42; and glucose incorporation into glycogen and increased glycogen content if hearts also infused with acetoacetate: Russell RR, et al. *JCI* 1997;100:2892). Some specific suggestions for revising this work:

Thank you for this assessment, which is actually slightly embarrassing to me (RDE), as I did my PhD in the same Krebs' lab (Metabolic Research Lab, Oxford) with Reg Hems, though after Heinrich had left. However I know Heinrich well and greatly admire and respect his work, including the BJ paper of 1980 which he quotes above; I feel very guilty I didn't quote it in the original submission I'm really sorry we didn't give due acknowledgment of his work on ketones-carbohydrates/glycogen, and, but I have attempted to do so now. We now quote the original BJ 1980 paper from the MRL, and the additional paper from Russell & Taegtmeyer's subsequent work with acetoacetate in his lab. My sincere apologies, and I hope the new version addresses and acknowledges this pioneering work.

So, I absolutely agree that Heinrich's earlier work with Reg and HAK and subsequently with Raymond Russell demonstrated that cardiac ketone body oxidation required anaplerosis, but (one could argue...) they did not definitively show that this was glycogen – they used glucose/pyruvate (OK, smoking gun &c&c, and clearly glycogen was a likely candidate but they didn't actually show that – as you say, they rescued cardiac function with glucose/pyruvate, not glycogen); nor did they measure anaplerosis. Furthermore you COULD argue that the timeline for their hearts' loss of function due to cataplerosis was not strictly in line with total glycogen depletion, although this is obviously very difficult to define. Whatever, it is certainly true that our M/S did not pay sufficient due to this critical early pioneering work (for which as I say I apologise) and I have attempted to correct this also in the text, which I hope is acceptable.

Regarding the issue of novelty with this work, I would make several points:

1. It is true that glycogen has been implicated as a myocardial anaplerotic resource, but this has never been definitively demonstrated, at least we would argue not. We would suggest that our work comes closest to demonstrating this proposed link mechanistically.
2. We have studied energetics by MR as well as cardiac function to better assess the metabolic, as well as mechanical function of the heart in relation to cataplerosis-anaplerosis. Again, we would argue that this is a novel approach and provides detailed information on energetic status during metabolic stress/challenge
3. We have employed metabolomics, again for the first time in this experimental context, as a means of further uncovering the underlying metabolic mechanism(s). We believe that the metabolomics data greatly enhances and supports the mechanistic conclusions.
4. Perhaps most critically, we have underplayed our entirely new technique to assess actual anaplerosis. (I did consider submitting this paper as a Methods paper and perhaps should have done so.) Measuring anaplerosis, as the Referees will certainly know, is a very difficult undertaking – Christine des Rosiers in Montreal has done so with stable radiolabelling but we believe this technique would be extremely challenging in the context of glycogen manipulation. Hence we came up with the method described here. We believe it has wide application – and indeed we are preparing a second M/S now looking at other aspects of cardiac anaplerosis.

We probably should have made greater play of the novelty of this experimental approach in the original M/S but completely understand why that (rather understated) assertion was lost in the first submission. I have modified the M/S somewhat to underline novelty issues but generally I have never really believed that the M/S is the place to trumpet novelty – hence I am pleased to try to underline the novelty aspect to Editors and Referees here. I do hope this makes sense.

- 1) The protocol to increase or decrease cardiac glycogen levels involving substrate-free perfusion vs glucose, insulin, pyruvate, and lactate somewhat complicates

some of these analyses as surely increased glycogen storage is not the only effect of prolonged exposure to no vs surplus metabolic substrates (for example, different nucleotide redox ratios?). Perhaps another way to perform a similar analysis would be to have all hearts with normal/similar glycogen content, but then infuse hearts with a glycogen phosphorylase inhibitor to prevent glycogenolysis.

Thank you for this important point. Yes, we agree that pre-perfusing hearts with a substrate-free perfusate may cause changes other than purely on glycogen. However, we did analyse a sub-set of hearts very carefully when we were establishing the technique (freeze-clamp them for metabolic analysis) but did not do MR energetics on them. We would argue that ALL interventions in a complex ex vivo system such as organ perfusion will potentially have multiple consequences, including glycogen phosphorylase inhibitors. Furthermore, once the experimental perfusion commenced – ie the actual study – there were no pharmacological effectors still present. In our defence I would say that as well as analysing these hearts (admittedly with the primary aim of checking [glycogen content]) we did use only a very brief substrate-free perfusion period, and we did find that they functioned well (Table 1). Arguably a bigger concern is that the glycogen stripping (substrate-free) per-perfusion period was shorter than the substrate-rich glycogen enhancing pre-perfusion – but we would argue that metabolic and functional data suggest that the two groups remained comparable.

2) In the perfusate to create high glycogen, it seems strange that pyruvate/lactate is provided at a 4:1 ratio. Typically lactate/pyruvate is provided at nearly 10:1 ratio to better mimic in vivo concentrations and better maintain redox ratios.

Again, thank you for an interesting point. Well, the choice of ratio is to some extent empirical, and of course the lactate:pyruvate ratio can vary widely physiologically. We would argue 1. That 10:1 is not grossly pathological, but does occur physiologically 2. We actually monitored the cytosolic redox state via lactate/pyruvate and perhaps most importantly 3. The high substrate initial loading conditions to augment tissue [glycogen] were only used during the pre-perfusion period, the idea being to provide an abundance of ALL substrates to facilitate and stimulate glycogen synthesis; in the actual experimental perfusion, more physiological substrate concentrations were deliberately chosen (though of course these can always be questioned – see below).

3) Perhaps they are also not different, but it would be beneficial to display both the heart rates and developed pressures in addition to RPP in Table 1.

Well this turned out to be an excellent suggestion as a couple of small differences did turn up (one in HR, one in DP) – see amended Table 1. They are now explicit

and discussed in the text, though we would argue they are not of great practical significance (though I suppose would lend some ammunition to those who say that RPP is too crude a metric of cardiac mechanical function – as might be seen clinically but is generally acceptable in animal work *ex vivo*). But we would stick to our original assertion that RPP is the accepted measure of cardiac mechanical function in the isolated perfused heart model and no significant differences were found.

4) The figure legends appear to provide the total number of animals used in the experiment and not the n per each group, which needs to be corrected.

Many thanks. All the Figures have now been re-drawn according to J Physiol guidelines at the Editor's request with individual data points included (so n is explicit) and means \pm SD) as required. Legends have been re-written to include exact P values. In addition, to aid clarity, I have included statistical marks (* # &c) on the Figures to aid their rapid and convenient observation (I see from recent J Physiol papers that this seems acceptable and it seems to me very helpful, given the large amount of data when all statistical analyses are quoted). In addition we have completed the Statistical Summary Document as required with all the data summarised. I hope this is all helpful but would be happy to further modify if required.

5) The quantified glycogen concentrations appear very high for the heart in my experience, even for the "low glycogen" group. For mouse hearts, glycogen is typically under 10 μ mol glycosyl/g wet weight, but perhaps there is a large difference between mouse and rat heart glycogen that I am not aware of. Conversely, is it possible that these units are actually per gram dry weight instead of wet weight? It would also be somewhat interesting to see "normal" rat heart glycogen concentrations here without any infusions altering glycogen abundance.

We don't have much experience with mouse hearts, but I would expect all rodent hearts to be fairly comparable. Our results are definitely per gm wet wt. I think the reason for the discrepancy is likely the assay technique (Bergmeyer?) – and this underlines the difficulty of cross-paper comparisons and (on a broader semi-philosophical note!) the essential requirement for within-group controls. In the event, we found the glycogen assay to be very accurate and reproducible. Given the ephemeral/rapidly changing nature of glycogen, we were struck at how reliable the estimates were and how consistent the measurements were within groups – the SDs aren't fantastic but for a highly variable substrate such as glycogen we think they're not bad. We have measured "normal" myocardial glycogen levels and found them to be entirely consistent with the levels we report here. So we are pretty confident with the amounts reported here.

6) Along the lines with the last comment above, there are some instances where statements are not technically correct. On page 11: "Initial perfusion with glucose-free buffer decreased myocardial glycogen"... this phrasing is not correct since it is not a pre vs post measurement but is instead comparing the glycogen content after no substrate perfusion vs high substrate perfusion. These types of statements are also made on page 14: "perfusion of high glycogen hearts with substrate-free buffer increased fumarate, which associated with decreased succinate". -Yes, those trends are true for comparing to perfusions with glucose and/or bHB, but the statement seems more of a comparison to what the heart content is before perfusing with substrate-free buffer. The wording just needs to be corrected here to compare the different substrate perfusions.

Yes, I'm sorry, the wording was shoddy in places. So I have (hopefully) tightened the wording. I can see now the Referee's issue with the wording in the text reporting the metabolomics (page 14 et seq.) – we actually chose to call these hearts "HG" thinking it would aid understanding but clearly it has caused confusion and I apologise for that. I have changed "HG" hearts to "glycogen-replete hearts" to emphasise that they have glycogen (ie aren't glycogen-stripped) but aren't strictly in the HG v LG groups. Hopefully this is clearer now?

Referee #2:

A. Kadir and colleagues have done a study in rat hearts investigating the importance of glycogen presence for substrate metabolism with special regards to glucose and ketone bodies.

The study is very well designed to answer the research question. I have only a few point.

Again, we are grateful for the Referee's careful appraisal of our work and kind comments.

Results:

Fig. 3C. In the legends it says N=42. To me it seems like only 10 hearts have been plotted? Could the authors explain?

Many thanks for this. Yes, we should have been more explicit with the n numbers (which refer to the number of individual observations ie the number of actual hearts perfused). In line with Editorial request, we have now included all data points on the Figures (and please see comment to Referee #1 comment 4) above).

It seems Pete Cox was a part of the study back in 2018, when it was presented as a poster: P31 Effect of glycogen content on ketone body oxidation and glycolysis in the isolated rat heart. How comes he is not an author in the final version?

Yes, indeed, thanks for this – Pete was involved principally with human research work on ketone esters in the lab and wasn't involved in rodent studies; he certainly helped with discussions on this work, certainly enough to justify inclusion in an Abstract, but probably was not sufficiently involved to merit inclusion on the full paper according to the Journal of Physiology's criteria for authors.

Could the authors maybe discuss the clinical impact of this finding in the paper. Should this be used a tool to decide whom would benefit from ketone bodies? Should glycogen build be attained?

Many thanks for this good suggestion. Yes, there is definitely a clinical angle on this work, especially in light of fairly recent evidence of the role of ketone bodies in cardiac pathological/failure states and the inter-relationship of metabolic disease and cardiac dysfunction. We don't really want to be too speculative in the manuscript but have added a short section highlighting this possibility, including the Bedi and Aubert references which are well known and referenced and really highlighted the issue of ketone body oxidation and heart disease a few years ago, which we hope is sufficient and acceptable.

ADDITIONAL FORMATTING REQUIREMENTS FOR RESUBMISSION:

-Include a Key Points list in the article itself, before the Abstract.

-Author photo and profile. First (or joint first) authors are asked to provide a short biography (no more than 100 words for one author or 150 words in total for joint first authors) and a portrait photograph. These should be uploaded and clearly labelled with the revised version of the manuscript. See Information for Authors for further details.

-You must start the Methods section with a paragraph headed Ethical Approval. A

detailed explanation of journal policy and regulations on animal experimentation is given in [<http://onlinelibrary.wiley.com/doi/10.1113/JP270818/full>]Principles and standards for reporting animal experiments in The Journal of Physiology and Experimental Physiology by David Grundy J Physiol, 593: 2547-2549. doi:10.1113/JP270818.). A checklist outlining these requirements and detailing the information that must be provided in the paper can be found at: <https://physoc.onlinelibrary.wiley.com/hub/animal-experiments>. Authors should confirm in their Methods section that their experiments were carried out according to the guidelines laid down by their institution's animal welfare committee, and conform to the principles and regulations as described in the Editorial by Grundy (2015). The Methods section must contain details of the anaesthetic regime: anaesthetic used, dose and route of administration and method of killing the experimental animals.

-The Reference List must be in Journal format

-Your manuscript must include a complete Additional Information section

-Please upload separate high-quality figure files via the submission form.

-Please ensure that the Article File you upload is a Word file.

-A Statistical Summary Document, summarising the statistics presented in the manuscript, is required upon revision. It must be on the Journal's template, which can be downloaded from the link in the Statistical Summary Document section here: https://jp.msubmit.net/cgi-bin/main.plex?form_type=display_requirements#statistics

-Papers must comply with the Statistics Policy https://jp.msubmit.net/cgi-bin/main.plex?form_type=display_requirements#statistics

In summary:

-If $n \leq 30$, all data points must be plotted in the figure in a way that reveals their range and distribution. A bar graph with data points overlaid, a box and whisker plot or a violin plot (preferably with data points included) are acceptable formats.

-If $n > 30$, then the entire raw dataset must be made available either as supporting information, or hosted on a not-for-profit repository e.g. FigShare, with access details provided in the manuscript.

-'n' clearly defined (e.g. x cells from y slices in z animals) in the Methods. Authors should be mindful of pseudoreplication.

-All relevant 'n' values must be clearly stated in the main text, figures and tables, and the Statistical Summary Document (required upon revision)

-The most appropriate summary statistic (e.g. mean or median and standard deviation) must be used. Standard Error of the Mean (SEM) alone is not permitted.

-Exact p values must be stated. Authors must not use 'greater than' or 'less than'. Exact p values must be stated to three significant figures even when 'no statistical significance' is claimed.

-Statistics Summary Document completed appropriately upon revision

-A Data Availability Statement is required for all papers reporting original data. This must be in the Additional Information section of the manuscript itself. It must have the paragraph heading "Data Availability Statement". All data supporting the results in the paper must be either: in the paper itself; uploaded as Supporting Information for Online Publication; or archived in an appropriate public repository. The statement needs to describe the availability or the absence of shared data. Authors must include in their Statement: a link to the repository they have used, or a statement that it is available as Supporting Information; reference the data in the appropriate section(s) of their manuscript; and cite the data they have shared in the References section. Whenever possible the scripts and other artefacts used to generate the analyses presented in the paper should also be publicly archived. If sharing data compromises ethical standards or legal requirements then authors are not expected to share it, but must note this in their Statement. For more information, see our Statistics Policy.

-Please include an Abstract Figure file, as well as the figure legend text within the main article file. The Abstract Figure is a piece of artwork designed to give readers an immediate understanding of the research and should summarise the main conclusions. If possible, the image should be easily 'readable' from left to right or top to bottom. It should show the physiological relevance of the manuscript so readers can assess the importance and content of its findings. Abstract Figures should not merely recapitulate other figures in the manuscript. Please try to keep the diagram as simple as possible and without superfluous information that may distract from the main conclusion(s). Abstract Figures must be provided by authors no later than the revised manuscript stage and should be uploaded as a separate file during online submission labelled as File Type 'Abstract Figure'. Please ensure that you include the figure legend in the main article file. All Abstract Figures should be created using BioRender. Authors should use The Journal's premium BioRender account to export high-resolution images. Details on how to use and access the premium account are included as part of this email.

Dear Dr Evans,

Re: JP-RP-2022-284270X "On the interdependence of ketone body oxidation, glycogen content, glycolysis and energy metabolism in the heart" by Rhys David Evans, Azrul Abdul Kadir, Brianna Stubbs, Cher-Rin Chong, Henry Lee, Mark Cole, Carolyn A Carr, David Hauton, James McCullagh, and Kieran Clarke

Thank you for submitting your revised Research Article to The Journal of Physiology. It has been assessed by the original Reviewing Editor and Referees and has been well received. Some final additional items have been requested. (See required items below.)

We hope you will be able to return your revised manuscript within one week. If you require longer than this, please contact journal staff: jp@physoc.org.

REVISION CHECKLIST:

""PLEASE NOTE - Please upload two versions of your manuscript text: one with all relevant changes highlighted and one clean version with no changes tracked. The manuscript file should include all tables and figure legends, but each figure/graph should be uploaded as separate, high-resolution files.

We look forward to receiving your revised submission.

Yours sincerely,

Michael C. Hogan
Senior Editor
The Journal of Physiology
<https://jp.msubmit.net>
<http://jp.physoc.org>
The Physiological Society
Hodgkin Huxley House
30 Farringdon Lane
London, EC1R 3AW
UK
<http://www.physoc.org>
<http://journals.physoc.org>

REQUIRED ITEMS FOR REVISION

-You must start the Methods section with a paragraph headed Ethical Approval. A detailed explanation of journal policy and regulations on animal experimentation is given in Principles and standards for reporting animal experiments in The Journal of Physiology and Experimental Physiology by David Grundy J Physiol, 593: 2547-2549. doi:10.1113/JP270818.). A checklist outlining these requirements and detailing the information that must be provided in the paper can be found at: <https://physoc.onlinelibrary.wiley.com/hub/animal-experiments>. Authors should confirm in their Methods section that their experiments were carried out according to the guidelines laid down by their institution's animal welfare committee, and conform to the principles and regulations as described in the Editorial by Grundy (2015). The Methods section must contain details of the anaesthetic regime: anaesthetic used, dose and route of administration and method of killing the experimental animals.

â€"The Journal of Physiology funds authors of provisionally accepted papers to use the premium BioRender site to create high resolution schematic figures. Follow this link and enter your details and the manuscript number to create and download figures. Upload these as the figure files for your revised submission. If you choose not to take up this offer we require figures to be of similar quality and resolution. If you are opting out of this service to authors, state this in the Comments section on the Detailed Information page of the submission form. The link provided should only be used for the purposes of this submission. Authors will be charged for figures created on this premium BioRender account if they are not related to this manuscript submission.

â€"Please upload separate high-quality figure files via the submission form.

EDITOR COMMENTS

Reviewing Editor:

Comments to the Author (Required):

No further comments

Nice work

REFEREE COMMENTS

Referee #1:

Thank you for adequately responding to my previous critiques. Just for the record, I do not believe I personally used the term

"novelty" in my assessment, and therefore, I am now satisfied that the historical literature setting the stage for these current studies is now more properly cited. I agree with you that the previous studies did not definitively show that glycogen was the source of anaplerosis, which you do in this current manuscript. I do think this is a very nice study and a great new technique for assessing cardiac anaplerosis.

Referee #2:

My comments have been met sufficiently.

END OF COMMENTS

1st Confidential Review

14-Dec-2022

Kadir et al

On the interdependence of ketone body oxidation, glycogen content, glycolysis and energy metabolism in the heart

Original J Physiol submission: JP-RP-2022-284017

Authors responses to acceptance email:

We are very grateful for the careful consideration given to our work, and kind words of support, and are very pleased that it has now been accepted for publication. I believe I have addressed the final few editorial requests, viz:

1. The Methods section starts with an Ethical Approval paragraph, and details of anaesthetic regime, as requested as per house style.
2. I have submitted a BioRender version of the Abstract Figure (or at least believe I have – my apologies, I am totally unfamiliar with this graphical package – the programme said it had downloaded). In addition I am submitting this Abstract Figure in the form of a Powerpoint slide (the means by which it was created) and a PNG image file. The regular Figures in the text are submitted as a Powerpoint file (I'm not sure if each Figure should be submitted separately?). I do hope this is all acceptable – I don't really have sufficient IT skill to know how to manipulate these images beyond what I have submitted, However, if further work is required please let me know and I will try again.
3. I have been through the checklist for submission and believe I have addressed all points; again, please get back to me if not.

Many thanks again for your consideration and agreeing to publish our work

Your sincerely

Rhys D Evans

Dear Dr Evans,

Re: JP-RP-2022-284270X "On the interdependence of ketone body oxidation, glycogen content, glycolysis and energy metabolism in the heart" by Rhys David Evans, Azrul Abdul Kadir, Brianna Stubbs, Cher-Rin Chong, Henry Lee, Mark Cole, Carolyn A Carr,

David Hauton, James McCullagh, and Kieran Clarke

Thank you for submitting your revised Research Article to The Journal of Physiology. It has been assessed by the original Reviewing Editor and Referees and has been well received. Some final additional items have been requested. (See required items below.)

We hope you will be able to return your revised manuscript within one week. If you require longer than this, please contact journal staff: jp@physoc.org.

Your revised manuscript should be submitted online using the link in your Author Tasks: <https://jp.msubmit.net/cgi-bin/main.plex?el=A2JS1FXZ4A3CbH3F5A9ftdXKvRY8is4vlwFC6KhGBaQZ>. This link is accessible via your account as Corresponding Author; it is not available to your co-authors. If this presents a problem, please contact journal staff (jp@physoc.org). Image files from the previous version are retained on the system. Please ensure you replace or remove any files that are being revised.

REVISION CHECKLIST:

""PLEASE NOTE - Please upload two versions of your manuscript text: one with all relevant changes highlighted and one clean version with no changes tracked. The manuscript file should include all tables and figure legends, but each figure/graph should be uploaded as separate, high-resolution files.

We look forward to receiving your revised submission.

Yours sincerely,

Michael C. Hogan
Senior Editor
The Journal of Physiology
<https://jp.msubmit.net>
<http://jp.physoc.org>

The Physiological Society
Hodgkin Huxley House
30 Farringdon Lane
London, EC1R 3AW
UK
<http://www.physoc.org>
<http://journals.physoc.org>

REQUIRED ITEMS FOR REVISION

-You must start the Methods section with a paragraph headed Ethical Approval. A detailed explanation of journal policy and regulations on animal experimentation is given in Principles and standards for reporting animal experiments in The Journal of Physiology and Experimental Physiology by David Grundy J Physiol, 593: 2547-2549. doi:10.1113/JP270818.). A checklist outlining these requirements and detailing the information that must be provided in the paper can be found at: <https://physoc.onlinelibrary.wiley.com/hub/animal-experiments>. Authors should confirm in their Methods section that their experiments were carried out according to the guidelines laid down by their institution's animal welfare committee, and conform to the principles and regulations as described in the Editorial by Grundy (2015). The Methods section must contain details of the anaesthetic regime: anaesthetic used, dose and route of administration and method of killing the experimental animals.

â€"The Journal of Physiology funds authors of provisionally accepted papers to use the premium BioRender site to create high resolution schematic figures. Follow this link and enter your details and the manuscript number to create and download figures. Upload these as the figure files for your revised submission. If you choose not to take up this offer we require figures to be of similar quality and resolution. If you are opting out of this service to authors, state this in the Comments section on the Detailed Information page of the submission form. The link provided should only be used for the purposes of this submission. Authors will be charged for figures created on this premium BioRender account if they are not related to this manuscript submission.

â€"Please upload separate high-quality figure files via the submission form.

EDITOR COMMENTS

Reviewing Editor:

Comments to the Author (Required):

No further comments

Nice work

REFEREE COMMENTS

Referee #1:

Thank you for adequately responding to my previous critiques. Just for the record, I do not believe I personally used the term "novelty" in my assessment, and therefore, I am now satisfied that the historical literature setting the stage for these current studies is now more properly cited. I agree with you that the previous studies did not definitively show that glycogen was the source of anaplerosis, which you do in this current manuscript. I do think this is a very nice study and a great new technique for assessing cardiac anaplerosis.

Referee #2:

My comments have been met sufficiently.

END OF COMMENTS

Dear Dr Kadir,

Re: JP-RP-2023-284270XR1 "On the interdependence of ketone body oxidation, glycogen content, glycolysis and energy metabolism in the heart" by Rhys David Evans, Azrul Abdul Kadir, Brianna Stubbs, Cher-Rin Chong, Henry Lee, Mark Cole, Carolyn A Carr, David Hauton, James McCullagh, and Kieran Clarke

We are pleased to tell you that your paper has been accepted for publication in The Journal of Physiology.

Authors should note that it is too late at this point to offer corrections prior to proofing. The accepted version will be published online, ahead of the copy edited and typeset version being made available. Major corrections at proof stage, such as changes to figures, will be referred to the Editors for approval before they can be incorporated. Only minor changes, such as to style and consistency, should be made at proof stage. Changes that need to be made after proof stage will usually require a formal correction notice.

Yours sincerely,

Michael C. Hogan
Senior Editor
The Journal of Physiology
<https://jp.msubmit.net>
<http://jp.physoc.org>
The Physiological Society
Hodgkin Huxley House
30 Farringdon Lane
London, EC1R 3AW
UK
<http://www.physoc.org>
<http://journals.physoc.org>

P.S. - You can help your research get the attention it deserves! Check out Wiley's free Promotion Guide for best-practice recommendations for promoting your work at www.wileyauthors.com/eeo/guide. You can learn more about Wiley Editing Services which offers professional video, design, and writing services to create shareable video abstracts, infographics, conference posters, lay summaries, and research news stories for your research at www.wileyauthors.com/eeo/promotion.

IMPORTANT NOTICE ABOUT OPEN ACCESS: To assist authors whose funding agencies mandate public access to published research findings sooner than 12 months after publication, The Journal of Physiology allows authors to pay an Open Access (OA) fee to have their papers made freely available immediately on publication.

You can check if your funder or institution has a Wiley Open Access Account here: <https://authorservices.wiley.com/author-resources/Journal-Authors/licensing-and-open-access/open-access/author-compliance-tool.html>.

EDITOR COMMENTS

Now acceptable.